# Strategies for synergistic reduction of plastic leakage and greenhouse gas emissions in China

Jingjing Bai[1], Zichun Huang[1], Xuewei Liu[2], Yuxin Liu[1], Lingyu Tai[1], Ziyang Lou ⓘ[3], Johann Fellner[4], Wei Liu[5] & Wenchao Ma ⓘ[1,2] ✉

The plastics industry is now confronting the intertwined challenges of environmental leakage and greenhouse gas emissions. Although policy interventions may exert synergistic reduction effects, the magnitude of such synergy remains underexplored. Here, we systematically analyze the material metabolism, environmental leakage, and greenhouse gas emissions associated with 14 plastic types in China over the period 1992–2021, and model the synergistic emissions reduction potentials and relative cost-effectiveness of these plastics under 14 scenarios between 2021 and 2060. Our results show significant heterogeneity in historical emission trajectories across plastic categories. By 2060, the system change scenario demonstrates the greatest potential for synergistic emission reductions and optimized cost-effectiveness. Relative to the 2060 baseline scenario, the system change achieves an 80% reduction in plastic leakage and a 63% decrease in greenhouse gas emissions. These results provide a reference for the development of synergistic emission reduction strategies suitable for different plastic types and industries.

China has been battling the environmental leakage of plastics for nearly three decades, and despite significant achievements, it still faces daunting challenges[1]. In 2020, China was still ranked fourth in the world in terms of environmental leakage of plastics[2]. China has seized the opportunity to fulfill its climate commitments under the Paris Agreement, but plastics-related greenhouse gas (GHG) emissions continue to soar year after year[3,4]. About 40% of global plastics-related GHG emissions are linked to China[5]. It is therefore crucial to enhance the sustainable transformation of the plastics industry and to ensure that policies to control the environmental leakage of plastics and reduce GHG emissions are effectively implemented. Investigating the co-benefits of environmental policies can reduce social costs while supporting their implementation. Synergistic reductions in environmental leakage of plastics and GHG emissions depend to a large extent on the use of different types of measures, but these co-benefits may not be realized for every measure. For example, while large-scale incineration can be effective in reducing environmental leakage of plastics, it may currently result in increased GHG emissions[6,7]. Together, we aim to reduce environmental leakage and GHG emissions from all types of plastics. This requires in-depth analyses of consumption, environmental leakage and GHG emissions of various types of plastics.

The granularity of plastic Material Flow Analysis (MFA) determines the accuracy in quantifying environmental leakage and GHG emissions, as it provides the dataset for subsequent assessments. Current studies have constructed a framework for analyzing the material flow of specific plastics at the global[8], regional (e.g., China[9], U.S.[10], Japan[11], etc.), or industry levels (packaging[12], medical[13], express delivery[14], etc.). However, due to the complexity and diversity of plastic types, most existing

[1]Key Laboratory of Agro-Forestry Environmental Processes and Ecological Regulation of Hainan Province, School of Environmental Science and Engineering, Hainan University, Haikou, China. [2]School of Environmental Science and Engineering, Tianjin University, Tianjin, China. [3]School of Environmental Science and Engineering, Shanghai Engineering Research Center of Solid Waste Treatment and Resource Recovery/ China Institute for Urban Governance, Shanghai Jiao Tong University, Shanghai, China. [4]Institute for Water Quality and Resource Management, TU Wien, Vienna, Austria. [5]Department of Environmental Science and Engineering, Nankai University Binhai College, Tianjin, China. ✉e-mail: mawc916@tju.edu.cn

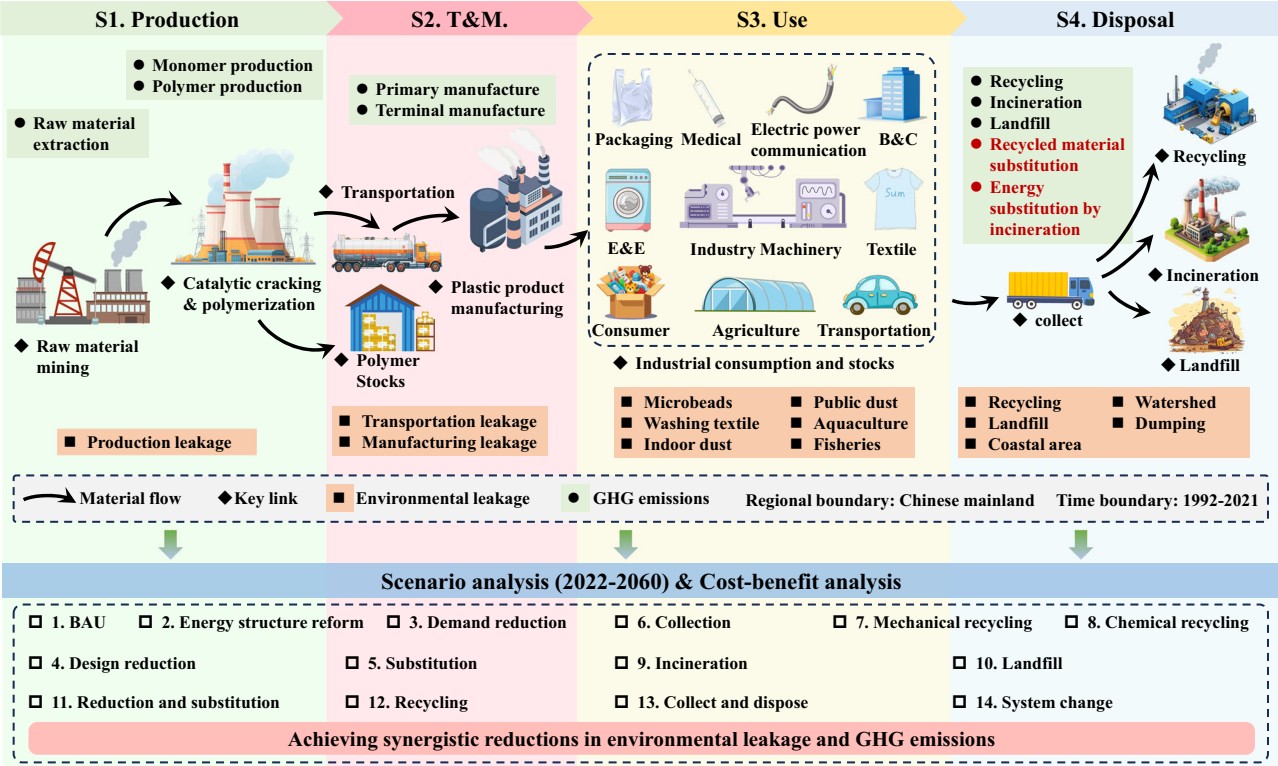

**Fig. 1 | System boundaries and technical routes of the coupled MFA-Leakage-GHG model.** This figure delineates the research framework for quantifying synergistic reduction potentials of 14 major polymers in China. The model encompasses four life-cycle stages: S1 (Stage 1)−Production, involving feedstock extraction and polymerization; S2 (Stage 2)−Transport and Manufacturing (T&M); S3 (Stage 3)−Use, covering consumption across ten sectors such as packaging, textiles, and electronics; and S4 (Stage 4)−Disposal, characterizing waste management pathways like recycling, incineration, and landfilling. The methodology integrates Material Flow Analysis (MFA) with environmental leakage accounting (macro- and micro-plastics) and life-cycle greenhouse gas (GHG) emission modeling, incorporating carbon offsets from material substitution and energy recovery. The bottom section presents the scenario analysis (2022−2060), evaluating 10 single and 4 composite measures based on their synergistic efficacy and cost-effectiveness. This framework provides a quantitative basis for China's plastic pollution and climate governance. Abbreviations: BAU business as usual, B&C Building and Construction, E&E Electrical and Electronics. The icons used in this figure were sourced from 588ku.com under the appropriate license.

studies have focused on 5-6 high-consumption categories, while there is still a large number of gaps in the material flow tracking of engineering plastics and plastics for special applications. This selective research perspective stems from the following major bottlenecks: on the one hand, insufficient data on petrochemical feedstock production leads to a lack of transparency in the upstream supply chain. On the other hand, differences in recycling rates at the plastic type-sector crossover weaken the reliability of data acquisition at the disposal stage. Particularly for low-consumption plastic types, the waste streams rely on empirical assumptions rather than field research data, further limiting the model's ability to generalize. Systematic deficiencies in the database make it difficult for existing material flow models to accurately track the dynamic migration of plastics through the supply chain, thus affecting the accuracy of synergistic assessment of environmental leakage and carbon emissions.

In the direction of quantifying plastic leakage, early studies have established a macro-estimation system for land-based plastic transport to the ocean based on coarse-grained modeling of socio-economic statistics and waste management parameters[15], but subsequent validation has shown that such methods can lead to systematic overestimation of evaluation results[16]. In addition, there is a lack of systematic quantification of plastics leakage in production and manufacturing processes, such as fugitive trimmings and microplastics by-products. In recent years, research perspectives have been expanded to include refined modeling of specific leakage pathways such as fishery activities[17], river transport[16,18], and coastal zone management[19], but there is still a modeling blind spot in terms of the coupling of

plastic-type coverage with the cascading leakage mechanism of the whole lifecycle stages of production-manufacturing-consumption-disposal[20,21]. For GHG emissions research related to plastics, the Life Cycle Assessment (LCA) methodology systematically analyzes the carbon emission distribution characteristics of the whole chain from fossil fuel extraction, polymer synthesis, product manufacturing to waste plastics disposal[7,22,23], and based on this, it deduces the technological feasibility of key emission reduction paths such as recycling economy transformation and bio-based material substitution[24,25]. However, the following core challenges remain: divergent system boundary delineation across studies, unknown mechanisms for coupling environmental leakage and GHG emissions, and a lag in updating non-mainstream plastics databases.

Current research needs to construct a coupled plastic leakage-carbon flow framework embedded with geographic region attributes to provide a theoretical breakthrough for the synergistic management of plastic pollution and climate change. In this study, a comprehensive dataset encompassing material metabolism, environmental leakage, and GHG emissions associated with plastics in China (Fig. 1) is constructed. This dataset spans multiple processes and covers the 14 most widely used polymers between 1992 and 2021. Subsequently, we analyzed the temporal trends in total environmental leakage and net GHG emissions for each type of plastic, and their contribution to total plastic pollution in China. Finally, the potential and relative cost-effectiveness of achieving synergistic emission reductions under different scenarios are explored, taking into account key factors such as the types of plastics and the characteristics of their consumption.

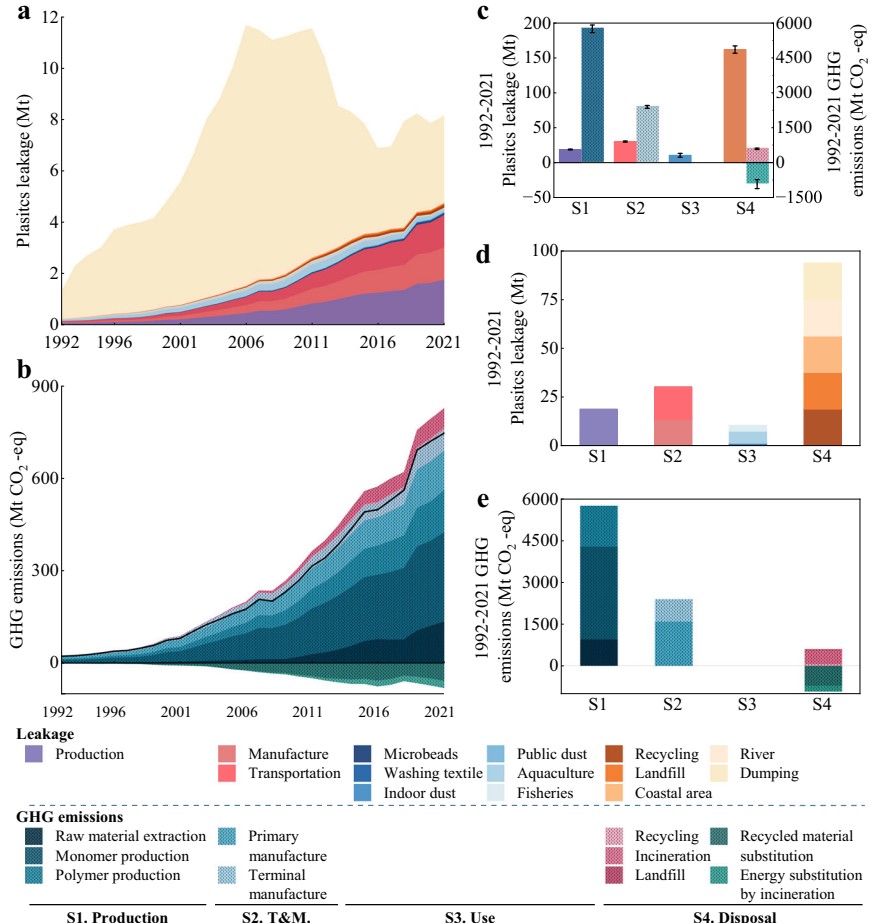

**Fig. 2 | Environmental leakage and associated GHG emissions from plastics in China, 1992–2021.** **a** Annual environmental leakage from 14 pathways in four stages. **b** Annual GHG emissions from 10 pathways in four stages. The black line indicates net GHG emissions. **c** Cumulative environmental leakage of plastics and associated GHG emissions in China from 1992 to 2021, divided into four stages: S1 (Stage 1)−Production; S2 (Stage 2)−Transport and Manufacturing (T&M); S3 (Stage 3)−Use; and S4 (Stage 4)−Disposal. Corresponds to the partition in the illustration. **d** Cumulative emissions from environmental leakage of plastics in China from 1992 to 2021, divided into 14 pathways. **e** Cumulative plastics-related net GHG emissions in China from 1992 to 2021, divided into 10 pathways. Uncertainty and sample size. For (**c**–**e**), bars show the median values across $n = 1000$ Monte Carlo simulations; error bars indicate the 95% uncertainty interval. One Monte Carlo run constitutes one independent sample.

## Results

### Plastic material flow analysis, environmental leakage and related GHG emissions

From 1992 to 2021, China's primary plastics production surged from 4.7 million metric tons (Mt) to 166.4 Mt, at a compound annual growth rate of 12.7% (Supplementary Fig. 4), exceeding the global average growth rate of 4%[8]. Considering "high imports, low exports" and stocks of primary plastics, and "low imports, high exports" of plastics products, China's consumption of plastics products has increased dramatically from 9.5 Mt to 106.0 Mt, with PET products seeing particularly significant growth.

Total environmental leakage increased rapidly from 1.3 Mt (Terrestrial, Ter. 1.2; Aquatic, Aq. 0.1) in 1992 to 11.7 Mt (Ter. 11.0, Aq. 0.7) in 2006, then slowly decreased to 6.9 Mt (Ter. 6.4, Aq. 0.5) in 2016, and finally increased to 8.2 Mt (Ter. 7.8, Aq. 0.4) in 2021(Fig. 2a). At the same time, the annual net GHG emissions increased from 21.9 Mt $CO_2$-eq (Emissions, Emiss. 22.5, Offset. 0.6) in 1992 to 679.1 Mt $CO_2$-eq (Emiss. 759.9, Offset. 80.8) in 2021(Fig. 2b). Among the four stages, production (Stage 1, S1) has the highest contribution to cumulative net GHG emissions (74%). It is worth noting that disposal (Stage 4, S4) shows a negative contribution (−5%) to cumulative net GHG emissions due to carbon offsetting effects from recycling and incineration processes. However, disposal (S4) was the most significant source (73%) of

cumulative environmental leakage (Fig. 2c). The results show that the structural characteristics of plastic leakage and GHG emissions show significant differences across life cycle stages and different emission pathways (Fig. 2d-e).

The main hotspots for environmental leakage of plastics in China were Polyethylene terephthalate (PET), Polypropylene (PP), Low-density polyethylene (LDPE), Polyvinyl chloride (PVC) and High-density polyethylene (HDPE) (Supplementary Fig. 5). In 1992, LDPE produced the largest volume of environmental leakage (0.4 Mt, 30%), followed by PP (0.3 Mt,19%) and Others (0.2 Mt, 18%), while PET and PVC accounted for only 6% and 5%, respectively. From 1992 to 2010, leakage of PET increased the most (3.5 Mt, 30%), followed by PVC (1.3 Mt, 11%) and PP (1.2 Mt, 10%). In 2006, emissions of a wide range of plastics reached a peak, mainly expanded polystyrene (EPS), PET, general purpose polystyrene (GPPS), and polycarbonate (PC). Since 1999, China has introduced various laws, regulations and policies related to "phasing out and banning disposable foamed plastic tableware"[26] and "initiating the development of China's circular economy"[27]. Other types of plastics peaking around 2010, with the implementation of policies such as the Plastic Restriction Order(2007)[28], the paid use of plastic bags(2008)[29], and the establishment of a comprehensive and advanced recycling system for used goods(2011)[30], there has been a significant

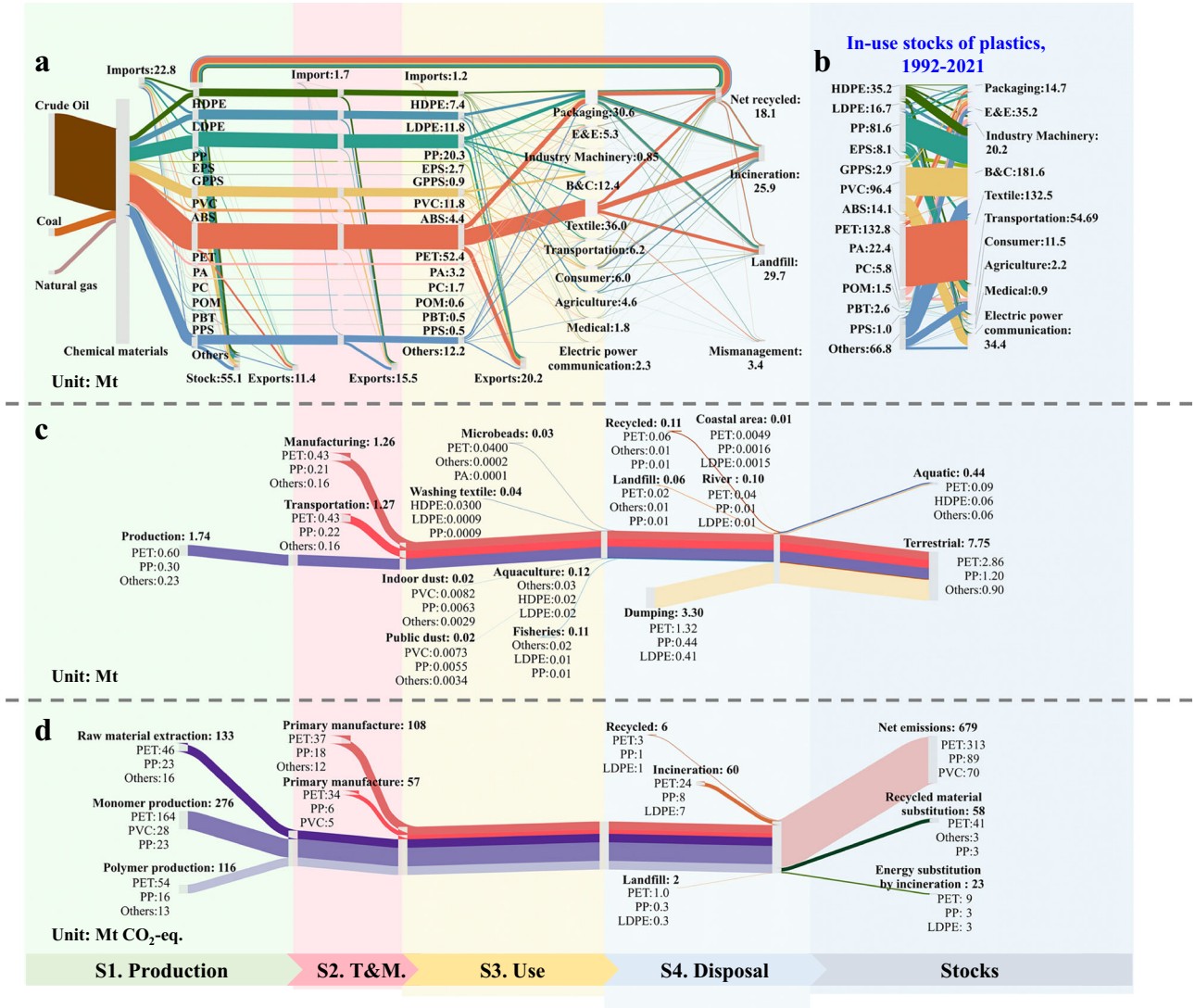

**Fig. 3 | Analysis of plastics material flows, environmental leakage, and GHG emissions in China in 2021. a** Plastics from fossil material to treatment and disposal in China in 2021. **b** In-use stocks of 14 types of plastics in 10 sectors, 1992–2021. **c** Environmental leakage accounting for 4 stages and 14 pathways for plastics in China in 2021. **d** GHG emissions accounting for 4 stages and 10 processes of plastics in China in 2021. Abbreviations: S1 (Stage 1)—Production, S2 (Stage 2)—Transport and Manufacturing (T&M), S3 (Stage 3)—Use, and S4 (Stage 4)—Disposal.

reduction in the environmental leakage of plastics such as PP, HDPE, and LDPE[31].

In contrast to the environmental leakage of plastics, GHG emissions show a continuously increasing trend, which is mainly influenced by the continuous growth of production and the increasing proportion of incineration. In 1992, PET contributed the highest net GHG emissions (8.2 Mt $CO_2$-eq, 34%), followed by PP (13%) and PVC (10%), with LDPE and HDPE accounting for 8% and 7%, respectively. From 1992 to 2021, the most significant increase in net emissions is for PET (305.0 Mt $CO_2$-eq), mainly due to the continued growth in demand from the packaging and textile industries[12]. The emissions for PP, LDPE, and HDPE experienced substantial growth due to the increased demand from the packaging industry. The net increase in emissions for PVC was 68.4 Mt $CO_2$-eq, primarily as a result of the rapid expansion of China's construction sector during this timeframe[32].

Figure 3a illustrates the whole process of plastics in China in 2021, from fossil material processing, and industrial consumption to final disposal. The results of the study indicate that crude oil accounts for 87% of China total primary plastics production in 2021, coal chemicals for 8%, and natural gas for 5% (Supplementary Fig. 2).

As a country "rich in coal, limited in oil and scarce in natural gas", China is now focusing on investing in coal-based chemical technologies to reduce its dependence on imported oil and gas[24,33]. China's consumption of plastic products is estimated at 106.0 Mt, of which 30.6 Mt is used for packaging and 36.0 Mt for textiles. Various types of polymers are used to meet China's demand, with PET, PP, PE and PVC together accounting for 82% of total demand. In 2021, China achieved a net materialized recycling volume of 18.1 Mt, with a materialized recycling rate as high as 31%, which is 1.74 times higher than the global average level of waste plastics recycling[8,24]. Considering the cross-use of various types of plastics across industries, the current high recycling volumes are largely attributable to the high recycling efficiencies of well-segregated polymer waste streams (such as PET bottles, transportation)[34,35]. However, recycling rates are generally lower for products consisting of mixed materials (Supplementary Fig. 6). Plastic stocks increased from 8.0 Mt in 1992 to 487.0 Mt in 2021, which is 4.6 times the amount of plastic consumed. Long-life products dominate the plastics stock[8]. From an industry perspective, construction and textiles accounting for 64.4% of the stock (Fig. 3b).

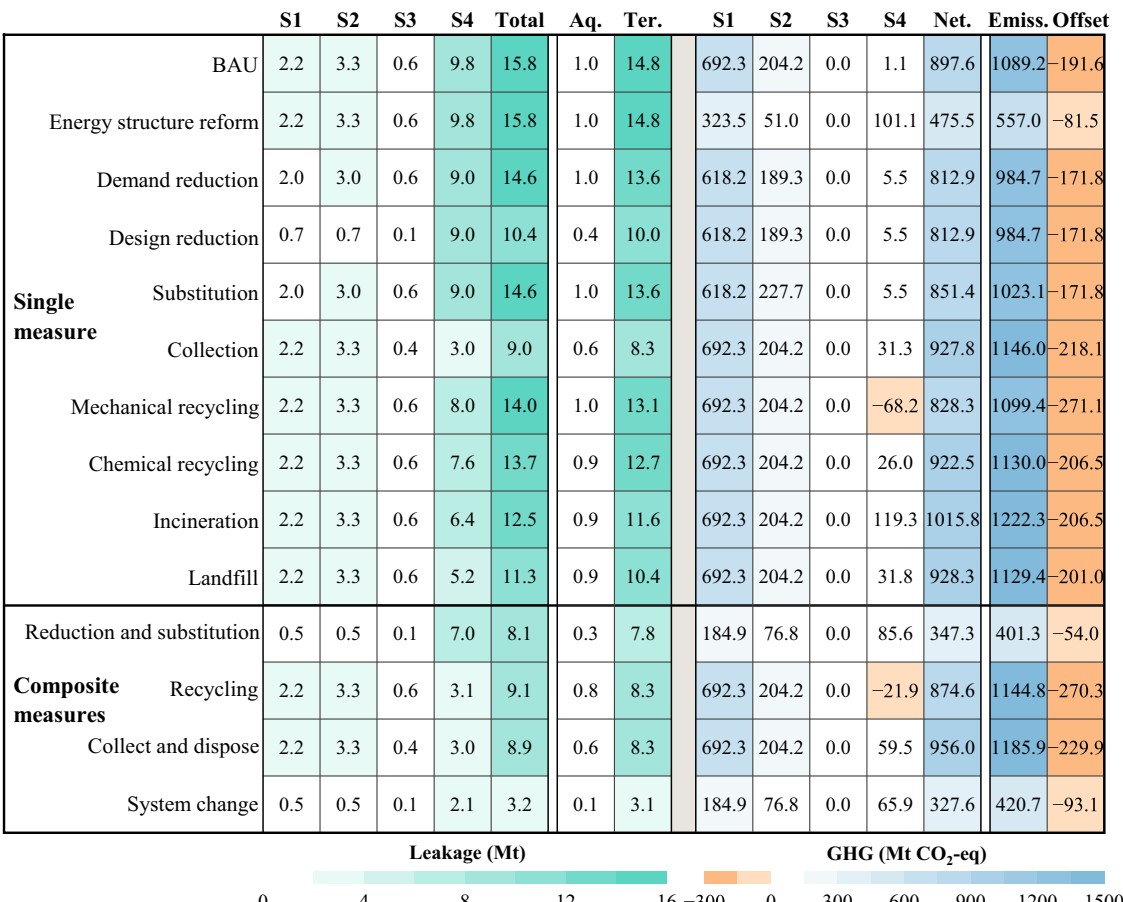

| | S1 | S2 | S3 | S4 | Total | Aq. | Ter. | | S1 | S2 | S3 | S4 | Net. | Emiss. | Offset |
|---|---|---|---|---|---|---|---|---|---|---|---|---|---|---|---|
| **Single measure** — BAU | 2.2 | 3.3 | 0.6 | 9.8 | 15.8 | 1.0 | 14.8 | | 692.3 | 204.2 | 0.0 | 1.1 | 897.6 | 1089.2 | −191.6 |
| Energy structure reform | 2.2 | 3.3 | 0.6 | 9.8 | 15.8 | 1.0 | 14.8 | | 323.5 | 51.0 | 0.0 | 101.1 | 475.5 | 557.0 | −81.5 |
| Demand reduction | 2.0 | 3.0 | 0.6 | 9.0 | 14.6 | 1.0 | 13.6 | | 618.2 | 189.3 | 0.0 | 5.5 | 812.9 | 984.7 | −171.8 |
| Design reduction | 0.7 | 0.7 | 0.1 | 9.0 | 10.4 | 0.4 | 10.0 | | 618.2 | 189.3 | 0.0 | 5.5 | 812.9 | 984.7 | −171.8 |
| Substitution | 2.0 | 3.0 | 0.6 | 9.0 | 14.6 | 1.0 | 13.6 | | 618.2 | 227.7 | 0.0 | 5.5 | 851.4 | 1023.1 | −171.8 |
| Collection | 2.2 | 3.3 | 0.4 | 3.0 | 9.0 | 0.6 | 8.3 | | 692.3 | 204.2 | 0.0 | 31.3 | 927.8 | 1146.0 | −218.1 |
| Mechanical recycling | 2.2 | 3.3 | 0.6 | 8.0 | 14.0 | 1.0 | 13.1 | | 692.3 | 204.2 | 0.0 | −68.2 | 828.3 | 1099.4 | −271.1 |
| Chemical recycling | 2.2 | 3.3 | 0.6 | 7.6 | 13.7 | 0.9 | 12.7 | | 692.3 | 204.2 | 0.0 | 26.0 | 922.5 | 1130.0 | −206.5 |
| Incineration | 2.2 | 3.3 | 0.6 | 6.4 | 12.5 | 0.9 | 11.6 | | 692.3 | 204.2 | 0.0 | 119.3 | 1015.8 | 1222.3 | −206.5 |
| Landfill | 2.2 | 3.3 | 0.6 | 5.2 | 11.3 | 0.9 | 10.4 | | 692.3 | 204.2 | 0.0 | 31.8 | 928.3 | 1129.4 | −201.0 |
| **Composite measures** — Reduction and substitution | 0.5 | 0.5 | 0.1 | 7.0 | 8.1 | 0.3 | 7.8 | | 184.9 | 76.8 | 0.0 | 85.6 | 347.3 | 401.3 | −54.0 |
| Recycling | 2.2 | 3.3 | 0.6 | 3.1 | 9.1 | 0.8 | 8.3 | | 692.3 | 204.2 | 0.0 | −21.9 | 874.6 | 1144.8 | −270.3 |
| Collect and dispose | 2.2 | 3.3 | 0.4 | 3.0 | 8.9 | 0.6 | 8.3 | | 692.3 | 204.2 | 0.0 | 59.5 | 956.0 | 1185.9 | −229.9 |
| System change | 0.5 | 0.5 | 0.1 | 2.1 | 3.2 | 0.1 | 3.1 | | 184.9 | 76.8 | 0.0 | 65.9 | 327.6 | 420.7 | −93.1 |

Leakage (Mt): 0 — 4 — 8 — 12 — 16  GHG (Mt CO₂-eq): −300 — 0 — 300 — 600 — 900 — 1200 — 1500

**Fig. 4 | Plastics environmental leakage and GHG emissions under different scenarios in 2060.** Total refers to the total amount of environmental leakage. Aq. refers to aquatic environment leakage. Ter. refers to terrestrial environmental leakage. Net. refers to net GHG emissions. Emiss. refers to total GHG emissions. Offset refers to GHG offsets from recycling substitution and incineration for electricity generation. Abbreviations: S1 (Stage 1)−Production, S2 (Stage 2)−Transport and Manufacturing (T&M), S3 (Stage 3)−Use, and S4 (Stage 4)−Disposal.

In 2021, PET accounted for the highest environmental leakage (2.9 Mt, 36%), followed by PP (15%), Others (12%), and PVC (10%), with LDPE and HDPE accounting for 10% and 7%, respectively (Fig. 3c). This is due to low recycling rates and poor management of PET fibers by the textile industry, which has led to significant leakage of PET plastics into the environment[36,37]. In four stages, disposal (S4) contributed the most to environmental leakage (44%), followed by Transport & Manufacturing, (Stage 2, S2, 31%), production (S1, 21%) and use (Stage 3, S3, 4%). Use (S3, 30%) and production (S1, 31%) were the main contributors to environmental leakage of microplastics. In S3, large quantities of microplastics are generated from personal care products containing microbeads[38], textile washing[39], house dust[40] and dust from public places. This is followed by disposal (S4, 15%) and the Transport & Manufacturing (S2, 11%). 0.17 Mt (63%) of microplastics enter the aquatic environment and 0.1 Mt (37%) enter the terrestrial environment. In terms of leakage of macroplastics, disposal (S4) has the highest leakage (45%), which is mainly due to insufficient management of end-of-life disposal[8,20]. This is followed by Transport & Manufacturing (S2, 32%), production (S1, 21%) and use (S3, 2%). With regard to GHG, production (S1) contributed the most to the cumulative emissions (69%) (Fig. 3d), followed by Transport & Manufacturing (S2, 22%) and disposal (S4, 9%), with the negative emissions from S4 offsetting about 11%.

Overall, different types of plastics show different trends and characteristics in terms of environmental leakage and GHG emissions, and these differences reflect the different applications of plastics in production and consumption, recycling rates and the effects of policy interventions. The results suggest that tailored mitigation strategies should be developed for each type of plastic.

## Scenario analysis of future environmental leakage and GHG emissions

In exploring effective measures to reduce plastic pollution, we have found that they broadly target specific stages of the life cycle. Specifically, scenarios such as demand reduction and energy structure reform (2–5) correspond to S1 and S2, while scenarios such as collection and recycling (6–10) are associated with stages S3 and S4. In addition, we have assessed the potential for different combinations of scenarios (11–14) to achieve synergistic emission reductions (Table 1). Despite the continued growth in demand for plastics (Supplementary Fig. 9–11), our results demonstrate that targeted mitigation strategies can achieve substantial reductions in environmental leakage and GHG emissions.

Under the business as usual (BAU) scenario, China's environmental leakage of plastics is projected to reach 15.8 Mt by 2060 (Ter. 14.8, Aq. 1.0), along with net GHG emissions of 916.5 Mt CO₂-eq (Emiss. 1111.2, Offset 194.7) (Fig. 4). Compared to BAU, the energy restructuring scenario in the single scenarios would lead to a 47% reduction in GHG emissions. However, the scenario has no impact on environmental leakage. On the other hand, substitution scenario has the weakest performance in reducing GHG emissions, with a reduction of only 5%, as the reduction was partially offset by the fact that substitution products still emit a certain amount of GHG. Design reduction scenario performs well and has the highest synergies in reducing

| | S1 | S2 | S3 | S4 | Total | Aq. | Ter. | | S1 | S2 | S3 | S4 | Net. | Emiss. | Offset |
|---|---|---|---|---|---|---|---|---|---|---|---|---|---|---|---|
| HDPE | 0.1 | 0.0 | 0.0 | 0.1 | 0.3 | 0.0 | 0.3 | | 5.8 | 6.2 | 0.0 | 8.0 | 20.0 | 25.6 | −5.6 |
| LDPE | 0.1 | 0.0 | 0.0 | 0.2 | 0.4 | 0.0 | 0.4 | | 7.1 | 12.4 | 0.0 | 14.4 | 33.9 | 43.8 | −9.9 |
| PP | 0.0 | 0.1 | 0.0 | 0.3 | 0.4 | 0.0 | 0.4 | | 14.0 | 13.4 | 0.0 | 9.6 | 37.0 | 45.3 | −8.3 |
| EPS | 0.0 | 0.0 | 0.0 | 0.0 | 0.0 | 0.0 | 0.0 | | 0.8 | 1.1 | 0.0 | 0.3 | 2.1 | 2.4 | −0.3 |
| GPPS | 0.0 | 0.0 | 0.0 | 0.0 | 0.0 | 0.0 | 0.0 | | 1.0 | 0.6 | 0.0 | 0.4 | 1.9 | 2.4 | −0.5 |
| PVC | 0.0 | 0.1 | 0.0 | 0.3 | 0.4 | 0.0 | 0.3 | | 18.1 | 3.9 | 0.0 | 4.2 | 26.2 | 32.1 | −6.0 |
| ABS | 0.0 | 0.0 | 0.0 | 0.1 | 0.1 | 0.0 | 0.1 | | 5.6 | 1.9 | 0.0 | 2.6 | 10.1 | 13.6 | −3.4 |
| PET | 0.1 | 0.2 | 0.1 | 0.7 | 1.1 | 0.1 | 1.0 | | 91.9 | 25.8 | 0.0 | 17.8 | 135.4 | 173.9 | −38.5 |
| PA | 0.0 | 0.0 | 0.0 | 0.1 | 0.1 | 0.0 | 0.1 | | 19.5 | 1.9 | 0.0 | −1.2 | 20.2 | 26.2 | −6.0 |
| PC | 0.0 | 0.0 | 0.0 | 0.0 | 0.0 | 0.0 | 0.0 | | 0.9 | 0.8 | 0.0 | 0.2 | 1.9 | 2.7 | −0.8 |
| POM | 0.0 | 0.0 | 0.0 | 0.0 | 0.0 | 0.0 | 0.0 | | 0.3 | 0.2 | 0.0 | 0.2 | 0.6 | 0.8 | −0.2 |
| PBT | 0.0 | 0.0 | 0.0 | 0.0 | 0.0 | 0.0 | 0.0 | | 1.4 | 0.2 | 0.0 | 0.3 | 1.9 | 2.4 | −0.5 |
| PPS | 0.0 | 0.0 | 0.0 | 0.0 | 0.0 | 0.0 | 0.0 | | 1.1 | 0.3 | 0.0 | 0.5 | 1.9 | 2.5 | −0.6 |
| Others | 0.1 | 0.1 | 0.0 | 0.3 | 0.5 | 0.0 | 0.4 | | 17.7 | 8.1 | 0.0 | 8.7 | 34.5 | 47.0 | −12.5 |

Leakage (Mt)                                                          GHG (Mt CO$_2$-eq)

0.0   0.2   0.4   0.6   0.8   1.0   1.2   −50   0   50   100   150   200

**Fig. 5 | Environmental leakage and GHG emissions of 14 plastics in 2060 under systems change scenario.** Total refers to the total amount of environmental leakage. Aq. refers to aquatic environment leakage. Ter. refers to terrestrial environmental leakage. Net. refers to net GHG emissions. Emiss. refers to total GHG emissions. Offset refers to GHG offsets from recycling substitution and incineration for electricity generation. Abbreviations: S1 (Stage 1)−Production, S2 (Stage 2)−Transport and Manufacturing (T&M), S3 (Stage 3)−Use, and S4 (Stage 4)−Disposal.

environmental leakage compared to substitution and demand reduction. Design reduction scenario results in a 9% reduction in GHG emissions and a 34% reduction in environmental leakage. This is made possible by factors such as lightweight design and reduced leakage rates during manufacturing included in the initiative, which improves environmental performance across the board. The implementation of collection scenario significantly reduced environmental spills by 43%. However, except for mechanical recycling, GHG emissions under all other scenarios in the disposal chain are on an increasing trend. Incineration will lead to a 13% increase in GHG emissions because the processes of recycling, incineration and landfilling of plastic waste all consume energy, release decomposition gases and emit GHG directly during the incineration process. In short, no single intervention is sufficient to solve the plastic problem.

Combined scenarios have better abatement potential than single scenarios, with environmental leakage reduction impacts ranging from 42 to 80% and GHG reduction potential from −6 to −63%. The integration of a full chain of emission reduction strategies, covering all stages to maximize their application potential, represents the most proactive response under current technological conditions: the system Change Scenario. With this strategy, synergistic reductions in environmental leakage and GHG emissions can be maximized.

With the implementation of the system change scenario, it is expected that the total amount of plastic leakage to the environment can be reduced to 3.2 Mt (Ter. 3.1, Aq. 0.1) by 2060 (Fig. 4), which is a reduction of up to 80% compared with BAU. At the same time, net GHG emissions can be reduced to 327.6 Mt CO$_2$-eq (Emiss. 420.7, Offset. 93.1), a reduction of 63%.

All types of plastics show varying degrees of effectiveness in reducing emissions, with environmental leakage ranging from 73% to 91% and net GHG reduction impacts ranging from 54% to 79% percent (Fig. 5, Supplementary Fig. 12). The reduction impacts of HDPE, LDPE, PP, EPS and GPPS are significantly higher than those of other plastic types. The collection rate of PET plastics is still low due to the extensive use of polyester fibers in the textile industry and the difficulties associated with recycling[41]. Therefore, the development of economically viable solutions for the efficient management of polyester fibers in the textile industry is essential to address the problem of plastic pollution.

In terms of the composition ratio of plastics emission reductions, PET, PP, Others, LDPE, HDPE and PVC are the major emission reduction plastics, accounting for 90% (environmental leakage) and 87% (GHG emissions) of the total emission reductions, respectively. These plastic types are expected to remain the main source of environmental leakage and GHG emissions until 2060, a phenomenon largely determined by their low economic value, dispersed sources of generation, difficulty of recycling, and large consumption base. Therefore, more attention and research should be given to treatment and emission reduction strategies for these plastic types.

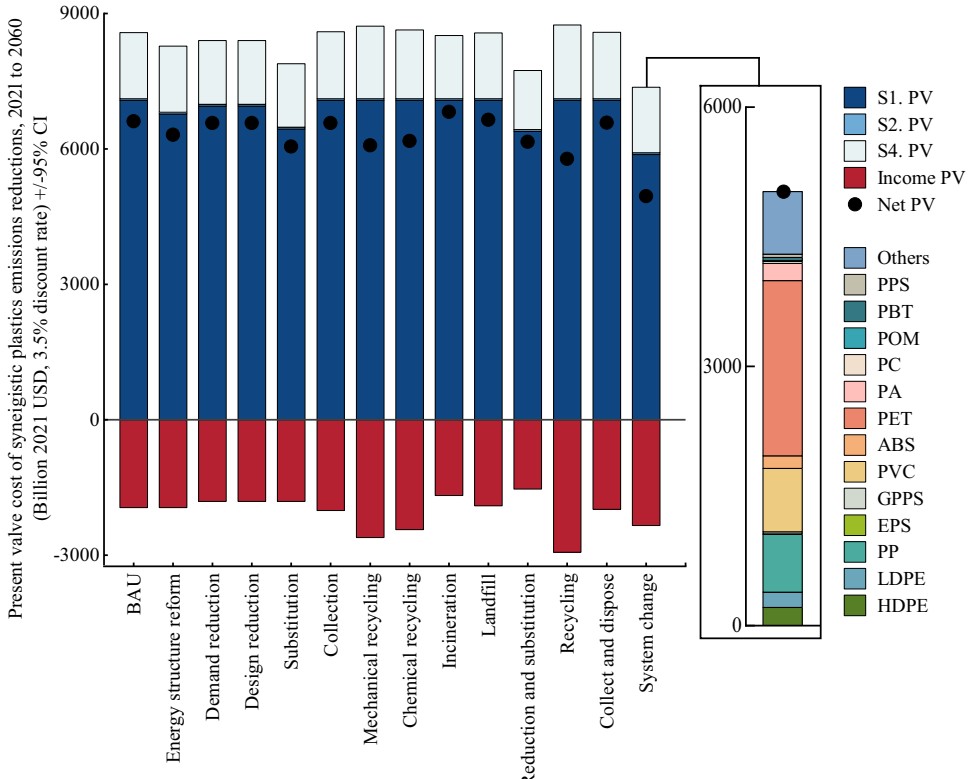

**Fig. 6 | Present value (PV) costs of plastics in China under different synergistic emission reduction scenarios.** The costs are calculated at a discount rate of 3.5% (0–7%), and income PV represents the benefits associated with the sale of recycled plastic materials and electricity generated from plastics incineration and energy recovery. The black dots represent the net benefits of the PV costs. In particular, we show the PV cost of each plastic under the system change scenario.

## Economic evaluation of synergistic emission reductions from plastics

The cumulative present value of synergistic plastics emission reductions in China from 2021 to 2060 was estimated to assess the relative cost-effectiveness of each scenario (Fig. 6). Among the scenarios, costs do not change by more than 25% relative to the BAU scenario, with incineration being the most costly of the single scenarios and collection and disposal being the most costly of the combined scenarios.

Compared to BAU, the system change scenario achieved a cost reduction of 24.8%, demonstrating optimum cost-effectiveness. This is due to the reduction in the number of plastics produced and the optimization of the energy mix, which significantly reduces costs in the production chain. In addition, the benefits of reduced consumption, recycling substitution and incineration further reduce costs. PET remains the main type in the expenditures, followed by Others, PVC and PP, the production constitutes the main stage of cost expenditures. This study provides a valuable reference for policymakers to identify the key types and stages of cost expenditures, which helps to formulate more precise and effective policies.

## Discussion

By integrating polymer-resolved material flow analysis with environmental leakage and GHG accounting, this study resolves the heterogeneous environmental profiles of plastics across life-cycle stages. Production, driven by fossil feedstocks and energy-intensive polymerization, dominates plastics-related GHG emissions[24], whereas end-of-life mismanagement governs environmental leakage[2,19,42]. Five polymers (PET, PP, LDPE, HDPE, and PVC) account for the majority of both impacts, indicating that environmental burdens are concentrated in specific material types and socioeconomic uses. These findings show that polymer properties and sectoral applications jointly determine environmental outcomes and underscore the need for differentiated mitigation strategies.

Scenario analysis indicates that single measures such as enhanced recycling or energy decarbonization are insufficient to generate substantial co-benefits. A mismatch exists between the drivers of upstream GHG emissions and the mechanisms of downstream plastic leakage. The system-change scenario integrates material reduction, design optimization, improved collection and recycling, and energy decarbonization, thereby reducing virgin demand at the source while lowering losses at end-of-life. By 2060, environmental leakage and net GHG emissions decline by roughly 80% and 63%, respectively. These results demonstrate that only coordinated, life-cycle-based governance can effectively deliver joint reductions in plastic pollution and climate impacts.

Differences across polymers and sectors further highlight the need for targeted mitigation. Plastic leakage is highly concentrated in a small set of critical material–sector combinations (Supplementary Tables 34–37). In aquatic systems, the dominant contributors are HDPE, LDPE, PP and textile PET; in terrestrial systems, leakage is primarily associated with packaging plastics (HDPE, LDPE, PP, PET), agricultural LDPE films and textile PET. These polymers exhibit both low circularity and high leakage risks, and their persistent dominance over time reveals structural weaknesses in product design, collection systems and end-of-life management. Targeted interventions for these high-risk polymers can generate disproportionate environmental benefits, align with carbon-neutrality and circular-economy objectives, and provide actionable pathways for differentiated commitments under an international plastics treaty.

China's in-use plastic carbon stock reached 1107.0 Mt $CO_2$-eq in 2021 and is projected to increase to 2485.0 Mt $CO_2$-eq by 2060 under the system-transition scenario (Supplementary Fig. 24). This growing

stock represents an increasingly significant pool of embedded carbon whose long-term accumulation and eventual release depend on product lifetimes, material substitution decisions, and the efficiency of recycling loops. As these dynamics are not captured by production-based accounting alone, integrating plastics management into national energy-transition strategies, circular manufacturing systems, and carbon-accounting frameworks is essential[25,43–45]. Such integration would allow China to track embodied-carbon dynamics more accurately, avoid future stock-driven emissions, and strengthen synergies between plastic governance and carbon-neutrality goals.

Overall, plastic mitigation requires differentiated and coordinated life-cycle strategies informed by polymer properties and sectoral applications. The polymer-resolved framework developed here provides the quantitative evidence of the coupled emission–leakage characteristics of China's plastic system, offering a scientific basis for synergistic management of plastic pollution and climate impacts. This framework can support global agreements in priority-setting, monitoring and differentiated implementation by identifying high-impact polymers, critical stages and optimal intervention pathways. Nationally, prioritizing high-contribution polymers, improving end-of-life collection and integrating circular material substitution, industrial decarbonization and carbon-stock accounting are expected to deliver the greatest combined benefits.

## Methods

### Constructing a dynamic flow map of plastics in China

Data on plastic production, end-use applications, and waste generation are dispersed, owing to the lack of a unified publication or official statistical authority that systematically compiles information across the entire plastic supply chain. Acquiring detailed and granular data on individual polymers at different stages of the plastic supply chain presents significant challenges. Nonetheless, such data are crucial for comprehending the types of polymers used in plastic products, predicting the service life of products in their applications, and determining when products transition into waste. To delineate the flow map of plastics, the primary task is to collate and integrate a multitude of disparate data sources. We traced upstream to dissect the material flow through the chemical processes from fossil feedstocks to primary plastics, thereby refining the material metabolism of plastics (Supplementary Fig. 2). Subsequently, we further organized data pertaining to the production, import and export, primary manufacturing, and consumption of 14 primary plastics in China, including HDPE, LDPE, PP, EPS, GPPS, PVC, Acrylonitrile Butadiene Styrene (ABS), PET, Polyamide (PA), PC, Polyoxymethylene (POM), Polybutylene Terephthalate (PBT), Polyphenylene Sulfide (PPS), and Others, with detailed information presented in Supplementary Table 2.

To analyze the import and export of plastics contained within finished goods, we examined trade data. These data are provided in physical units and encompass the production and trade of various products, yielding a total of 86,946 data records. For each product category, we estimated the polymer composition and its proportion by mass to construct a comprehensive overview of the production, import, and export totals for each product category and polymer type. Each product category was classified into one of the following stages of the supply chain: primary plastic form, intermediates, end-use manufactured goods, and waste.

The Dynamic Material Flow Analysis has been successfully applied in several studies to dynamically model inventories of materials, including plastics[46–49]. In this study, we adopt a similar approach, starting from the previously described flow map of plastics in China and disaggregating it by end-use product categories. For more details, please refer to Supplementary Method 1.

Current mainstream treatments for waste plastics comprise mechanical recycling, incineration, and landfilling. Chemical recycling has yet to achieve industry-scaled implementation but was incorporated into our scenario analysis[19,50]. Open dumping and uncontrolled disposal are uniformly categorized as mismanagement[2,16].

The main data that support the findings of this study are as follows: (1) Data on the production, consumption, and trade of primary plastics in China from 1992 to 2021 can be sourced from the China Plastics Industry Yearbook[51]; (2) Trade data on primary plastic products, finished plastic products and plastic waste can be obtained from the online statistical data service platform of the General Administration of Customs of the People's Republic of China[52] and the United Nations Comtrade database[53]; (3) Parameters for environmental leakage and GHG emissions were derived from the literature, with primary sources detailed in the Supplementary Method 2 and 3; (4) Population estimates can be obtained from the "Forecast of Medium and Long Term Change Trend of Chinese Population" compiled by the China Population and Development Research Center and the United Nations Population Fund China Representative Office[54]; (5) Estimations of future per capita Gross Domestic Product (GDP) are derived from the literature, with the primary sources detailed in the Supplementary Method 5. The mass balance of primary plastics is calculated according to Eq. (1):

$$C_{k,t} = PR_{k,t} - S_{k,t} + I_{k,t} - E_{k,t} \tag{1}$$

where $C_{k,t}$ is the consumption of primary plastics $k$ in year $t$, $PR_{k,t}$, $I_{k,t}$ and $E_{k,t}$ are the domestic production, import and export of primary plastics $k$ in year $t$, respectively, $S_{k,t}$ refers to the stock of primary plastics $k$ in year $t$.

The domestic consumption of products is determined according to Eq. (2):

$$C''_{k,p,t} = C_{k,t} * \alpha_{k,p,t} * (1 - \gamma) + I'_{k,p,t} - E'_{k,p,t} + I''_{k,p,t} - E''_{k,p,t} \tag{2}$$

where $C''_{k,p,t}$ is the domestic consumption of product $p$ produced by primary plastic $k$ in year $t$, $\alpha_{k,p,t}$ is the product diversion ratio, which indicates the proportion of primary plastic $k$ flowing into product $p$ in year $t$, and $\gamma$ is the scrap rate in the manufacturing process, $I'_{k,p,t}$ and $E'_{k,p,t}$ refer to the imports and exports of product $p$ produced by primary plastic $k$ in year $t$, $I''_{k,p,t}$ and $E''_{k,p,t}$ are the imports and exports of plastic $k$ into sector $p$ in year $t$, where it is produced.

The annual mass of plastic waste generated is estimated through the failure rate of each group of end-use products aged $t$ in product category $p$, characterized by a log-normal distribution[8,34,55]. For further details, refer to Supplementary Method 1. The waste generation is calculated according to Eq. (3):

$$OF_{k,p,t} = \sum_P \sum_{t=1}^{\infty} S_{n-1,t} \frac{\frac{1}{\sqrt{2\pi}Sx} e^{-\frac{(lnx-M)^2}{2S^2}}}{1 - \varphi(\frac{lnx-M}{S})} \tag{3}$$

where $OF_{k,p,t}$ is the outflow of plastic $k$ from sector $p$ in year $t$. The outflow is the waste generated by the sector. The risk rate for a log-normal distribution can be interpreted as the instantaneous failure probability for age $t$. This risk function is then used in the inventory model to calculate the share of each category of plastic products of age $t$ disposed of in any given year. where $\varphi$ is the cumulative distribution function for the standard normal $N(\mu = 0, \sigma^2 = 1)$.

### Estimation of plastic environmental leakage in China

Through a literature review[15,16,19,20,39,56–58], this study identifies the main pathways and stages of environmental leakage of plastics. Specific equations and parameter values are given below. For more details, please refer to Supplementary Method 2. Plastic leakage is calculated

according to Eq. (4), Eq. (5), and Eq. (6):

$$Leakage_{total,k,t} = Leakage_{ma,k,t} + Leakage_{mj,k,t} \qquad (4)$$

where $Leakage_{total,k,t}$ refers to the total environmental leakage of type $k$ plastics in year $t$, $Leakage_{ma,k,t}$ refers to the environmental leakage of macro-plastics of type $k$ plastics in year $t$, $Leakage_{mj,k,t}$ refers to the environmental leakage of micro-plastics of type $k$ plastics in year $t$.

$$Leakage_{ma,k,t} = Terr_{Ma,k,t} + Aqu_{Ma,k,t} \qquad (5)$$

where $Terr_{Ma,k,t}$ refers to the macroplastic leakage of type $k$ plastics into the terrestrial environment in year $t$, $Aqu_{Ma,k,t}$ refers to the macroplastic leakage of type $k$ plastics into the aquatic environment in year $t$.

$$Leakage_{mi,k,t} = Terr_{Mi,k,t} + Aqu_{Mi,k,t} \qquad (6)$$

where $Terr_{Mi,k,t}$ refers to the microplastic leakage of type $k$ plastics to the terrestrial environment in year $t$, $Aqu_{Mi,k,t}$ refers to the microplastic leakage of type $k$ plastics to the aquatic environment in year $t$.

### Estimation of plastics-related GHG emissions in China

This study establishes a carbon accounting system for the entire life cycle of plastics, centered on 10 segments: extraction of fossil raw materials, monomer processing, polymerization, primary manufacturing, terminal manufacturing, recycling, incineration, landfill, recycled material substitution and energy substitution by incineration[3,25]. For more details, please refer to Supplementary

Method 3. The total GHG emissions are estimated using Eq. (7):

$$GHG_{total,k,t} = ME_{k,t} + MP_{k,t} + Poly_{k,t} + PM_{k,t} + TM_{k,t} + Rec_{k,t} \\ + Inc_{k,t} + Landfill_{k,t} - RS_{k,t} - IS_{k,t} \qquad (7)$$

where $GHG_{total,k,t}$ is the total amount of GHG emissions associated with type $k$ plastics in year $t$, $ME_{k,t}$ is the GHG emissions associated with the extraction of feedstock for type $k$ plastics in year $t$, $MP_{k,t}$ is the GHG emissions associated with the processing of a single plastic of type $k$ in year $t$, $Poly_{k,t}$ is the polymerization-related GHG emissions of type $k$ plastics in year $t$, $PM_{k,t}$ is the primary processing-related GHG emissions of type $k$ plastics in year $t$, $TM_{k,t}$ is the end-process-related GHG emissions of type $k$ plastics in year $t$, $Rec_{k,t}$ is the GHG emissions associated with the recycling of type $k$ plastics in year $t$, $Inc_{k,t}$ is the GHG emissions associated with the incineration of type $k$ plastics in year $t$, $Landfill_{k,t}$ is the landfill-related GHG emissions of type $k$ plastics in year $t$, $RS_{k,t}$ is the GHG emissions reductions associated with the recycling substitution of type $k$ plastics in year $t$, $IS_{k,t}$ is the GHG emissions reductions from incineration substitution of type $k$ plastics in year $t$.

### Scenario simulation

Emissions scenario modeling is carried out in three main steps. First, we projected per capita plastic stocks based on GDP per capita, which we combined with population projections to derive annual stock levels of plastic types in the industry. Subsequently, we retrospectively calculate annual production, consumption, disposal, and stock levels for the BAU. Finally, we estimate environmental leakage and GHG emissions based on the above projections of future plastic material flows. It is worth noting that we

**Table 1 | Specifications of 10 single scenarios and 4 composite scenarios**

| | Scenario | By 2060 |
|---|---|---|
| | BAU | None |
| Single measure | Reform of the energy structure | 1. Energy consumption in the power system: hydroelectricity (25%), coal (15%), nuclear (15%), wind (25%), photovoltaic (20%); Industrial steam production: coal (15%), natural gas (30%), solid waste (0.55%); Fuel consumption structure: coal (10%), natural gas (10%), solid waste (15%), primary energy and others (65%). |
| | Design optimization for reduction | 2. The packaging sector targets 30% source reduction through design optimization, while prohibiting production of plastic-microbead household chemicals and PE mulch films below 0.01 mm thickness. |
| | Reduction in demand | 3. Demand in the packaging industry decreased by 30%. |
| | Reduction through substitution | 4. Five packaging substitutes (bamboo, wood, paper, glass, biodegradable plastics) equally share a 30% market substitution. |
| | Collection | 5. Mismanaged waste incidence reduced to 1%. |
| | Mechanical recycling | 6. Mechanical recycling achieves 50%. |
| | Chemical recycling | 7. Chemical recycling achieves 20%. |
| | Incineration | 8. Incineration achieves 60%. |
| | Landfilling | 9. Landfilling achieves 30%. |
| Composite measures | Reduction and substitution (1–4) | 10. Reform of the energy structure. The packaging sector targets 90% source reduction, while prohibiting production of plastic-microbead household chemicals and PE mulch films below 0.01 mm thickness. |
| | Recycling (6–7) | 11. Mechanical recycling achieves 35%. Chemical recycling achieves 25%. |
| | Collect and dispose (5, 8–9) | 12. Mismanaged waste incidence reduced to 1%. Incineration achieves 50%. Landfilling achieves 19%. |
| | System change (1–9) | 13. Reform of the energy structure. The packaging sector targets 90% source reduction, while prohibiting production of plastic-microbead household chemicals and PE mulch films below 0.01 mm thickness. Mismanaged waste incidence reduced to 1%. Mechanical recycling achieves 30%. Chemical recycling achieves 18%. Incineration achieves 30%. Landfilling achieves 21%. |

assume that net imports as a share of production or consumption in each phase remain constant as in 2021.

Based on the historical analysis, 10 key factors affecting environmental leakage and associated GHG emissions were identified within the definition (Table 1, Supplementary Table 28). The future scenario analysis was categorized into 10 single scenarios and 4 composite scenarios. When modifying a single scenario variable, scenario weightings remain constant under baseline assumptions. We have also developed scenarios for different plastic types in order to analyze their synergistic emission reduction potential. For more details, please refer to Supplementary Method 5 and 6.

## Cost and benefits

Plastics-related expenditures and revenues are the main influencing factors in predicting the net economic cost of coordinated reductions in plastic pollution in China. We used a discount rate of 3.5% (0–7%) to calculate the total net cost per year under different measures between 2022 and 2060. Detailed projections of the corresponding management expenditures and revenues were made based on the consumption of front-end plastics and the share of plastic waste collected, sorted, disposed of, recycled and incinerated under different future scenarios. Recycled plastics are processed into regenerated pellets that substitute virgin plastic feedstocks, thereby reducing upstream production costs in the industrial chain[19,59]. Incineration generates direct economic revenue through energy recovery (electricity/heat generation)[42,60]. Therefore, based on the projected plastic consumption and plastic waste emissions, the net economic costs of plastic waste management in China can be projected for different future scenarios from 2022 to 2060 by subtracting plastic-related revenues from waste management cost expenditures. For more details, please refer to Supplementary Method 7.

## Uncertainty analysis

In material flow analysis, Monte Carlo analysis is an effective tool for assessing the impact of uncertainty in model input parameters on model prediction results[61–64]. In plastic material flow analysis, the range of uncertainty for each input variable is determined based on the quality and origin of the data (i.e., data source). These uncertainties naturally propagate throughout the Monte Carlo simulation process, thus affecting the predicted results of plastic flow and mass change. To obtain more accurate predictions, 1000 Monte Carlo simulations of the plastic flow process were performed. For more information, see Supplementary Method 8.

## Reporting summary

Further information on research design is available in the Nature Portfolio Reporting Summary linked to this article.

## Data availability

The data generated in this study are provided in the Supplementary Information and figshare repository [https://doi.org/10.6084/m9.figshare.29875109][65].

## Code availability

The code developed in this study is available via figshare repository [https://doi.org/10.6084/m9.figshare.29875109][65].

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

## Acknowledgements
The authors gratefully acknowledge the financial support from NSFC-UNEP Joint Research Project with No. 72261147460 (W.M.), Hainan key R&D program with No. ZDYF2025SHFZO62 (W.M.).

## Author contributions
J.B. and W.M. co-designed the study. Z.H., X.L., Y.L., and L.T. contributed to data collection and processing, Z.L., J.F., and W.L. conducted technical analyses and results interpretation. J.B. wrote the paper and W.M. revised the paper.

## Competing interests
The authors declare no competing interests.
