## [Transparent Peer Review file · Nature Communications]

Strategies for synergistic reduction of plastic leakage and greenhouse gas emissions in China

Corresponding Author: Dr Wenchao Ma

Attachments originally included by the reviewers as part of their assessment can be found at the end of this file

Version 0:

Reviewer comments:

Reviewer #1

(Remarks to the Author)

Please see comments in the attached file.

Reviewer #2

(Remarks to the Author)

The authors have analyzed and quantified different strategies to minimize plastic leakage and GHG emissions from the production and disposal of plastics in China. The manuscript is robust and reads well with a solid structure. Following are some of my comments regarding the manuscript.

1. Line 24. What do you mean by material metabolism? Degradation of plastics? Please use the appropriate terminology wherever necessary throughout the manuscript. In Line 98 you use substance metabolism, and are these terms the same?
2. Line 31-32. Please rephrase the sentence. It reads as if plastic pollution is reduced by leakage compared to 2060 BAU.
3. Line 55-57 Please rephrase this sentence. I can't understand what the authors are trying to imply here
4. Line 67 What do you mean by low-volume plastic types? Is it based on the production volume or waste generation volume?
5. Line 84-87 Why parallel to the study of GHG emissions. LCA can also quantify the GHG emissions in the form of Global Warming Potential/Carbon Footprint, can't they?
6. Line 91 - Unknown mechanisms for coupling environmental leakage. What mechanism did the author use to incorporate the environmental leakage data and results within the framework of your study?
7. It will be ideal to concisely define the different EoL pathways considered in this manuscript so that the readers can understand the differences between EoL options like dumping, landfill or disposal
8. Line 124 What do you mean by metabolic map of plastics actually mean? Please rephrase or define the sentence
9. Line 139 Primary plastic form as in polymers? Also please provide examples for what intermediates exactly mean
10. Line 168 Estimation of plastic environmental leakage in China. How do you differentiate between macro and microplastics, and from which source you got the data to calculate such leakage values?
11. For the ISkt in the equation 7, what kind of substitution was considered for incineration? Is it just energy in general? Or electricity? Or electricity and heat? How much of is substituted in reality in China (BAU)?
12. Line 218 Please mention in the manuscript exactly what these different single and composite measures, even if you have mentioned them in the SI (section 7 and 8). Because, as a reader they need to know the specific measure for reducing the GHG emissions and leakage. Quantify the measures concisely in a table!!
13. Line 232-233 Recycling and incineration can generate revenue is a generic statement. Either explain it a bit or quantify them citing the relevant source
14. Line 236 'by removing the revenues..' why?
15. Line 245 1000 Monte Carlo simulations. What parameters or what kind of data were run in Monte-Carlo analysis and why? Due to the uncertainty of data?
16. Line 253, 255, 260 - Please cite the websites appropriately
17. Figure 2 need to be redone completely. Colour codes are confusing. Similar colours for incineration, landfill in the S4. Disposal. How do you differentiate between landfill and dumping? Because sometime landfill can be uncontrolled. What do you mean by Terminal manufacture? Why collection and sorting is not a part in Disposal? Please clarify
18. Line 292 Long-life products with textiles. Is it really the case? Any source to substantiate these claims?

19. If you could define/quantify the measures for different scenarios you can discuss them better in the Discussion section
20. I didn't understand Fig 6. Why there is a separate graph for different polymer types?

Reviewer #3

(Remarks to the Author)
Annotated manuscript attached

Version 1:

Reviewer comments:

Reviewer #1

(Remarks to the Author)
The manuscript presents valuable analytical findings; however, a significant weakness lies in the disconnection between these results and the policy implications section. The current implications are generic and not uniquely derived from the study's specific contributions. To strengthen the manuscript's impact, the authors must thoroughly reorganize this section to build a direct and actionable bridge from their results to concrete policy recommendations. Namely, authors should explicitly connect their results to actionable policy pathways. For instance, how can your current research findings be translated into practical applications to assist the government in managing plastics and carbon, as well as contribute to the development of an international plastics treaty? Does this study offer any enlightening insights for the management of international policies? How can these management policies be integrated with China's carbon neutrality goals to achieve synergistic emission reductions?

Reviewer #3

(Remarks to the Author)
Thank you for the careful review of the manuscript.

Version 2:

Reviewer comments:

Reviewer #1

(Remarks to the Author)
The paper has been revised well based on my previous comments, it is recommended to be accepted in its current form.

Itemized response to each review comment

“Strategies for synergistic reduction of plastic leakage and greenhouse gas emissions in China”

(NCOMMS-25-23856)

On behalf of my co-authors, we thank you very much for allowing us to revise our manuscript. We would like to express our sincere appreciation for your careful reading and the reviewers’ insightful comments concerning our manuscript. All the review comments have been adequately addressed and clearly explained in the revised version of the manuscript. All the changes in the manuscript have been shown with yellow highlighting. Here is the itemized response to each comment with a specific statement, clearly addressing each issue and explaining what, why and how of the corresponding revision.

Reviewer # 1

[Issue 1] The overall logic of the introduction is relatively clear, but the ending appears somewhat abrupt and could be improved. It is suggested to include a supplementary table comparing this study with previous research. A detailed comparison of methodologies, data sources, and conclusions could highlight the improvements in this study and describe how they were achieved.

[Response] Accepted. Thank you for your suggestion. We have added a supplementary table to systematize recent seminal research on global and Chinese plastic leakage and greenhouse gas (GHG) emissions. The table highlights the time span, methodological approach, core findings, and research lineage of the studies.

- **Supplementary Table 1. Literature findings and characteristics of plastic environmental leakage and GHG emissions in global and Chinese contexts**

Study area / Year of study	Value	Modeling & Methodology
-------	------------------------

Plastic leakage		
Plastic waste inputs from land into the ocean / 2010	9.6 / China: 2.425 ¹	Global terrestrial plastic marine leakage quantified by integrating solid waste, population density, and economic status datasets.
River plastic emissions to the world's oceans / 2015	1.78 ²	Developed a global riverine plastic emission model incorporating waste management, population density, and hydrology data.
Plastic waste leakage to the environment / 2016	30 ³	Assessed macroplastics from municipal sources and four terrestrial microplastic categories, focusing on environmental leakage pathways.
River plastic emissions to the world's oceans / 2017	0.4 ⁴	Constructed a global inventory of aquatic plastic debris across diverse river scales.
River plastic emissions to the world's oceans / 2018	0.13 ⁵	Developed a Human Development Index (HDI)-based predictive model for riverine plastic fluxes, calibrated and validated with field observations.
River plastic emissions to the world's oceans / 2020	1.0 / China: 0.071 ⁶	Integrated geospatial plastic waste, land use, wind, precipitation, and river data to quantify river-to-ocean transport probability.
Plastic waste leakage to the environment /2020	52.1/ China: 2.8 ⁷	Established a global macroplastic emissions inventory integrating emission mechanism modeling with empirical activity data.
Plastic waste inputs from land into the ocean / 2020	14.5 ⁸	Analysis of waste management data from 239 countries reveals significant infrastructure disparities between

		Global North/South, differentially impacting GHG emissions and marine plastic pollution.
Plastic waste leakage to the environment in China / 2020	China: 13.05 ⁹	Conducted dynamic MFA to quantify environmental plastic leakage in China.
River plastic emissions to the world's oceans / 2023	0.5 ¹⁰	Simulated riverine macro- and microplastic fluxes to coastal seas across >10,000 basins to identify dominant sources.
Plastic GHG emissions		
Global plastic GHG emissions / 2015	1700 ¹¹	Developed a dataset encompassing ten conventional and five bio-based plastics with lifecycle GHG emissions across multiple mitigation scenarios.
Global plastic GHG emissions / 2015	2000 / China: 820 (production perspective); 580 (consumption perspective) ¹²	Applied enhanced MRIO analysis to: quantify the global plastics carbon footprint (1995–2030) and fossil resource footprint across production lifecycles; map interregional trade linkages; project emissions under the IEA 2°C/6°C scenario; evaluate PM health impacts, employment, and value-added socioeconomic dimensions.
Global plastic GHG emissions / 2020	2200 ¹³	Models the complete plastic lifecycle: upstream chemical production, polymer synthesis, product manufacturing, sectoral applications, end-of-life management.
China plastics GHG emissions / 2020	496.8 ¹⁴	Assessed material flows of China's primary synthetic resins via MFA across production, manufacturing, and end-of-life stages; quantified stage-specific GHG emissions using standardized accounting.

Global plastic GHG emissions / 2050 (project)	3350 ¹⁵	Developed a machine learning integrated model projecting global plastic production, use, and end-of-life trajectories to 2050.
synergistic reduction of plastic leakage and greenhouse gas emissions		
This study /2021	Leakage: 8.2 (Ter. 7.8, Aq. 0.4) / GHG emissions:679.1	Quantifies the synergy potential of plastics in reducing environmental leakage and GHG emissions. Developed a systematic assessment framework analyzing historical leakage/carbon emissions for 14 polymers across 10 Chinese sectors. Pioneered co-benefit potential and cost-effectiveness quantification under 14 scenarios (2021-2060).

*Environmental leakage in million metric tons per year (Mt/yr); GHG emissions in million metric tons of CO₂-equivalent (Mt CO₂-eq).

[Issue 2] The fonts used in the figures should be standardized to ensure consistency. For example, in Figures 2 and 4, the font style makes certain words hard to read due to letters being too close together.

[Response] Accepted. We appreciate your suggestion. The following revisions have been implemented:

- Fig. 2: Split into main Fig. 2 and Supplementary Figure 7, with X-axis labels supplemented.
- Fig. 3: As requested by reviewer #2_question 4, add the comment box label in the upper right corner of 3a and change it to fig. 3b.
- Fig. 3-5: Unify font size and adjust word spacing.
- Fig. 6: Add net PV and adjust legend order.

Fig. 2 Environmental leakage and associated GHG emissions from plastics in China, 1992-2021. **a**, Annual environmental leakage from 14 pathways in four stages of plastics in China, 1992-2021. **b**, Annual GHG emissions from 10 pathways in four stages of plastics in China, 1992-2021. The black line indicates net GHG emissions, i.e., total GHG emissions minus total GHG offsets. Offsets include the substitution of raw materials for virgin plastics production by recovered recycled plastics and the substitution of carbon emissions from fossil energy electricity by plastics incineration for electricity generation. **c**, Cumulative environmental leakage of plastics and associated GHG emissions in China from 1992-2021, divided into four stages: Stage 1 (production),

Stage 2 (transport and manufacturing), Stage 3 (use) and Stage 4 (disposal). Corresponds to the partition in the illustration. **d**, Cumulative emissions from environmental leakage of plastics in China from 1992-2021, divided into 14 pathways. **e**, Cumulative plastics-related net GHG emissions in China from 1992 to 2021, divided into 10 pathways.

Supplementary Figure 7. Total macroplastic and microplastic leakage of plastics and their associated GHG emissions in China from 1992 to 2021. **a**, Microplastic leakage. **b**, Macroplastic leakage. **c**, Associated GHG emissions.

		S1	S2	S3	S4	Total	Aq.	Ter.	S1	S2	S3	S4	Net.	Emiss.	Offset
Single measure	BAU	2.2	3.3	0.6	9.8	15.8	1.0	14.8	692.3	204.2	0.0	1.1	897.6	1089.2	-191.6
	Energy structure reform	2.2	3.3	0.6	9.8	15.8	1.0	14.8	323.5	51.0	0.0	101.1	475.5	557.0	-81.5
	Demand reduction	2.0	3.0	0.6	9.0	14.6	1.0	13.6	618.2	189.3	0.0	5.5	812.9	984.7	-171.8
	Design reduction	0.7	0.7	0.1	9.0	10.4	0.4	10.0	618.2	189.3	0.0	5.5	812.9	984.7	-171.8
	Substitution	2.0	3.0	0.6	9.0	14.6	1.0	13.6	618.2	227.7	0.0	5.5	851.4	1023.1	-171.8
	Collection	2.2	3.3	0.4	3.0	9.0	0.6	8.3	692.3	204.2	0.0	31.3	927.8	1146.0	-218.1
	Mechanical recycling	2.2	3.3	0.6	8.0	14.0	1.0	13.1	692.3	204.2	0.0	-68.2	828.3	1099.4	-271.1
	Chemical recycling	2.2	3.3	0.6	7.6	13.7	0.9	12.7	692.3	204.2	0.0	26.0	922.5	1130.0	-206.5
	Incineration	2.2	3.3	0.6	6.4	12.5	0.9	11.6	692.3	204.2	0.0	119.3	1015.8	1222.3	-206.5
	Landfill	2.2	3.3	0.6	5.2	11.3	0.9	10.4	692.3	204.2	0.0	31.8	928.3	1129.4	-201.0
Composite measures	Reduction and substitution	0.5	0.5	0.1	7.0	8.1	0.3	7.8	184.9	76.8	0.0	85.6	347.3	401.3	-54.0
	Recycling	2.2	3.3	0.6	3.1	9.1	0.8	8.3	692.3	204.2	0.0	-21.9	874.6	1144.8	-270.3
	Collect and dispose	2.2	3.3	0.4	3.0	8.9	0.6	8.3	692.3	204.2	0.0	59.5	956.0	1185.9	-229.9
	System change	0.5	0.5	0.1	2.1	3.2	0.1	3.1	184.9	76.8	0.0	65.9	327.6	420.7	-93.1

Fig. 4 Plastics environmental leakage and GHG emissions under different scenarios in 2060.

S1, S2, S3 and S4 represent production, T&M, use and disposal, respectively. Total refers to the total amount of environmental leakage. Aq. refers to aquatic environmental leakage. Ter. refers to terrestrial environmental leakage. Net. refers to net GHG emissions. Emiss. refers to total GHG emissions. Offset refers to GHG offsets from recycling substitution and incineration for electricity generation.

Fig. 5 Environmental leakage and GHG emissions of 14 plastics in 2060 under systems change scenario.

Fig. 6 Present value (PV) costs of plastics in China under different synergistic emission reduction scenarios. The costs are calculated at a discount rate of 3.5% (0-7%) , and income PV represents the benefits associated with the sale of recycled plastic materials and electricity generated from plastics incineration and energy recovery. The black dots represent the net benefits of the PV costs. In particular, we show the PV cost of each plastic under the system change scenario.

[Issue 3] In Figure 2a, should the final label on the x-axis be revised to "2021" for consistency?

Additionally, are the x-axis labels missing in subfigures d, e, and f?

[Response] Accepted. We appreciate your suggestion. The timeline has been updated to 1992, 1996, 2001, 2006, 2011, 2016, 2021. Additionally, the original Fig. 2 has been split into current Fig. 2 and Supplementary Figure 7, with all X-axis labels supplemented.

[Issue 4] What is the significance of the small section on the right side of Figure 3a? Including a corresponding annotation or note would enhance the clarity of the figure.

[Response] Accepted. We appreciate you highlighting this issue. We acknowledge insufficient interpretation of this figure segment. The right-hand panel of Fig. 3a has been changed to Fig. 3b. This section shows the in-use stocks of plastic polymer types by sector. This is in contrast to the instantaneous flows of plastic materials in 2021 shown in Fig. 3a. Fig. 3b presents cumulative inventory dynamics, whose data magnitude is inherently different from the instantaneous flows.

Fig. 3 Analysis of plastics material flows, environmental leakage, and GHG emissions in China in 2021. a, Plastics from fossil material to treatment and disposal in China in 2021. b, In-use stocks of 14 types of plastics in 10 sectors, 1992-2021. c, Environmental leakage accounting for 4 stages and 14 pathways for plastics in China in 2021. d, GHG emissions accounting for 4 stages and 10 processes of plastics in China in 2021.

[Issue 5] What is the calculation method and basis for the step from chemical feedstock to primary plastics? Please provide a more detailed explanation.

[Response] **Acknowledged.** We appreciate this insightful question. Current plastic flow studies exhibit critical data gaps in tracing chemical transformations from fossil feedstocks to primary plastics – a pivotal nexus for accurately quantifying material flows and GHG emissions. Our research endeavors to establish a refined pathway bridging this gap. Establishing this refined pathway requires resolving three critical issues: **1. Mass ratios of chemical monomers for polymerization:** Quantify types and mass proportions of monomers required to synthesize specific polymers (e.g., mass of terephthalic acid and ethylene glycol needed per tonne of PET produced). **2. Transformation pathways of monomers:** Trace chemical conversion routes from base feedstocks to target monomers. **3. Proportionate contribution of fossil sources:** Determine the share of oil, coal, and natural gas in monomer production. Upon completing these steps, we established a comprehensive material flow pathway from fossil feedstocks to plastic polymers tailored to China's industrial context. As shown in Supplementary Figure 2.

- **Mass ratios of chemical monomers for polymerization:** Through systematic synthesis of scientific literature and chemical industry data, we quantified monomer mass ratios for 14 plastic polymers, with dedicated analysis of distinct production processes for identical polymers. Taking vinyl chloride monomer synthesis for polyvinyl chloride as an illustrative case, we focused on China's two dominant industrial routes: the carbide acetylene route and ethylene-based process.
- **Transformation pathways of monomers:** Employing a retrospective tracing approach, we clarified step-by-step transformation routes for each target monomer, ultimately linking them to primary feedstocks (e.g., naphtha, ethane, propane, methanol).
- **Proportionate contribution of fossil sources:** This phase establishes quantitative linkages between three fossil feedstocks (oil, coal, natural gas) and primary chemical feedstocks (e.g., naphtha, ethane, propane, methanol), with dedicated quantification of their consumption shares in China's chemical sector.

Supplementary Figure 2. Detailed flows of plastics from raw materials to primary plastics in China.

[Issue 6] In the section on plastic product production and use, could a table be added to clarify the conversion from plastic resins to primary products? The table could also match these products to the corresponding production sectors, listing the proportion of different products flowing into each sector, along with the source of the conversion coefficients.

[Response] Acknowledged. We appreciate your suggestion. Supplementary Table 2 delineates allocation ratios from plastic polymer types to primary plastic products. Furthermore, we have added Supplemental Table 3 documenting allocation ratios from primary plastic products to sectors alongside their literature sources.

[Issue 7] While the paper provides the composition ratios of various plastic resins in sectoral plastic products, it remains unclear how these data are used to calculate the volume of primary plastic products entering each sector.

[Response] Acknowledged. We appreciate your inquiry. The allocation from plastic polymer types to end-use sectors requires a two-step procedure:

- Plastic polymers are manufactured into primary products (e.g., pipes, panels). These primaries are usually standard sizes.

- These primary products are distributed to different industrial sectors and processed into products suitable for subsistence production. The complete datasets are documented in Supplementary Table 2 (polymer-to-product) and Supplementary Table 3 (product-to-sector).
- Industry consumption = China's primary plastics consumption × polymer to semi-finished product conversion factor × semi-finished product to industry distribution factor.

[Issue 8] Please review all formulas in the paper. Some formulas, such as Formula (6) in the main text and Formulas (10) and (11) in the appendix, contain variables or symbols not clearly linked to their explanations.

[Response] Accepted. We appreciate you highlighting these issues, which resulted from our oversight. We have thoroughly checked and revised all relevant formulas.

- **Line 460-467:**

$$Leakage_{ma,k,t} = Terr_{Ma,k,t} + Aqu_{Ma,k,t} \quad (5)$$

$Terr_{Ma,k,t}$ refers to the macroplastic leakage of type k plastics into the terrestrial environment in year t , $Aqu_{Ma,k,t}$ refers to the macroplastic leakage of type k plastics into the aquatic environment in year t .

$$Leakage_{mi,k,t} = Terr_{Mi,k,t} + Aqu_{Mi,k,t} \quad (6)$$

$Terr_{Mi,k,t}$ refers to the microplastic leakage of type k plastics to the terrestrial environment in year t , $Aqu_{Mi,k,t}$ refers to the microplastic leakage of type k plastics to the aquatic environment in year t .

- **Supplementary material Eqs.**

$$Leakage_{ma,k,t} = Terr_{Ma,k,t} + Aqu_{Ma,k,t} \quad (10)$$

$Terr_{Ma,k,t}$ refers to the macroplastic leakage of type k plastics into the terrestrial environment in year t , $Aqu_{Ma,k,t}$ refers to the macroplastic leakage of type k plastics into the aquatic environment in year t .

$$Leakage_{mi,k,t} = Terr_{Mi,k,t} + Aqu_{Mi,k,t} \quad (11)$$

$Terr_{Mi,k,t}$ refers to the microplastic leakage of type k plastics to the terrestrial environment in year t , $Aqu_{Mi,k,t}$ refers to the microplastic leakage of type k plastics to the aquatic environment in year t .

$$ME_{g,k,t} = Oil_{k,t} \times Oil_{p,t} \times Oil_{\beta,t} + Coal_{k,t} \times Coal_{p,t} \times Coal_{\beta,t} + NG_{k,t} \times NG_{p,t} \times NG_{\beta,t} \quad (20)$$

$Oil_{k,t}$, $Coal_{k,t}$, $NG_{k,t}$ refers to the amount of crude oil, coal and natural gas feedstock required for type k plastics in year t, $Oil_{p,t}$, $Coal_{p,t}$, $NG_{p,t}$ refers to the share of domestic production of crude oil, coal, and natural gas in total consumption in year t (Supplementary Table 13), $Oil_{\beta,t}$, $Coal_{\beta,t}$, $NG_{\beta,t}$ Refers to GHG emissions per unit of product for crude oil, coal, and natural gas in year t (Supplementary Table 14).

[Issue 9] Several coefficients used in the calculations, along with their sources, are not clearly explained. It is recommended to include a supplementary table listing these coefficients—for example, the coefficients for plastic leakage to the environment, and emission factors for greenhouse gases during recycling, incineration, and landfilling.

[Response] Accepted. We appreciate your suggestion. We acknowledge insufficient documentation of this information and have now systematically supplemented the relevant content in the Supplementary Materials.

● Supplementary Table 13. Sources of the data for model inputs.

	Plastic Life Stages	Leakage direction	Leakage Sources	Leakage Parameters	Sources	
Macro-plastic	$Prod_{Ma,\alpha}$	Terrestrial	Total	1.000%	Luan, X. et al ⁹	
	$Manuf_{Ma,\alpha}$	Terrestrial	Total	0.750%	Luan, X. et al ⁹	
	$Transp_{Ma,\alpha}$	Terrestrial	Total	0.750%	Luan, X. et al ⁹	
	Use	$Aquac.ind_{k,t}$	Aquatic	-	-	Bai, M. et al ³⁷ , Luan, X. et al ⁹
		$Fish.ind_{k,t}$	Aquatic	-	-	Bai, M. et al ³⁷ , Luan, X. et al ⁹
	$Rec_{Ma,\alpha}$	Terrestrial	Recycling	0.400%	Luan, X. et al ⁹	
	$Landfill_{Ma,\alpha}$	Terrestrial	Landfill	0.100%	Luan, X. et al ⁹	
	$Coastal_{ma,\alpha}$	Aquatic	Mismanagement	1-5%	Strokal, M. et al ¹⁰ , Lau, W. W. Y. et al ³	
	$River_{ma,\alpha}$	Aquatic	Mismanagement	-	Strokal, M. et al ¹⁰	
$Misman_{Ma,k,t}$	Terrestrial	-	-	-		

Micro-plastic		$Prod_{Mi,\alpha}$	Aquatic	Total	0.050%	Luan, X. et al ⁹	
		$Manuf_{Mi,\alpha}$	Aquatic	Total	0.005%	Luan, X. et al ⁹	
		$Transp_{Mi,\alpha}$	Terrestrial	Total	0.002%	Luan, X. et al ⁹	
	Use	$PCP_{k,t}$	Aquatic	-	-	-	Strokal, M. et al ¹⁰ , Lau, W. W. Y. et al ³
		Laundry Products	Aquatic	Textiles consumption	0.120%	0.120%	Strokal, M. et al ¹⁰ , Lau, W. W. Y. et al ³
		Indoor Dust	Terrestrial	-	0.080%	0.080%	Strokal, M. et al ¹⁰ , Luan, X. et al ⁹
		Public Dust	Terrestrial	-	0.080%	0.080%	Luan, X. et al ⁹
		$Rec_{Mi,\alpha}$	Aquatic	Recycling	0.030%	0.030%	Lau, W. W. Y. et al ³ , Luan, X. et al ⁹
	$Landfill_{Mi,\alpha}$	Terrestrial	Landfill	0.120%	0.120%	Lau, W. W. Y. et al ³ , Luan, X. et al ⁹	

- Supplementary Table 23. Parameters for greenhouse gas emissions from plastic waste recycling (t CO₂ -eq/t)^{11,29}.

disposal technology	Recycling
GHG emissions	0.906

- Supplementary Table 24. Parameters for greenhouse gas emissions from plastic waste incineration (t CO₂ -eq/t)^{11,29}.

disposal technology	Incineration
GHG emissions	2.351

- Supplementary Table 25. Parameters for greenhouse gas emissions from plastic waste incineration (t CO₂ -eq/t)^{11,29}.

disposal technology	Landfill
GHG emissions	0.089

[Issue 10] In supplementary formula (20), how is $Oil_{k,t}$ calculated? Also, should the references to Table S11 and Table S12 in the explanation be corrected to TableS12 and Table S13? Please check for similar errors throughout the text.

[Response] Accepted. We appreciate your correction of this error. We have modified the information in the supplementary table referenced in Equation 20. As new tables have been added to the Supplementary Materials, all referenced table numbers in this section have been revised to correspond to Supplementary Table 14 and Supplementary Table 15.

- **Supplementary material _ page 43:** $Oil_{k,t}$, $Coal_{k,t}$, $NG_{k,t}$ refers to the amount of crude oil, coal and natural gas feedstock required for type k plastics in year t, $Oil_{p,t}$, $Coal_{p,t}$, $NG_{p,t}$ refers to the share of domestic production of crude oil, coal, and natural gas in total consumption in year t (Supplementary Table 14), $Oil_{\beta,t}$, $Coal_{\beta,t}$, $NG_{\beta,t}$ Refers to GHG emissions per unit of product for crude oil, coal, and natural gas in year t (Supplementary Table 15).
- $Oil_{k,t}$ denotes the consumption of petroleum feedstocks required for producing plastic polymer k in year t. As addressed in Issue 5, we first established a comprehensive material flow pathway from fossil feedstocks to plastic polymers tailored to China's industrial context (Supplementary Fig. 2). Subsequently, we quantified the consumption coefficients of target monomers, primary feedstocks, and fossil energy inputs per unit of plastic polymer produced. By integrating actual production data of plastic polymers, the total fossil feedstock consumption for plastic production was derived, which explicitly includes petroleum inputs $Oil_{k,t}$.

Reviewer # 2

[Issue 1] Line 24. What do you mean by material metabolism? Degradation of plastics? Please use the appropriate terminology wherever necessary throughout the manuscript. In Line 98 you use substance metabolism, and are these terms the same?

[Response] **Accepted.** Thank you for your constructive feedback. We acknowledge having previously neglected the distinction between "Material Metabolism" and "Substance Metabolism," and have now standardized the terminology to "**Material Metabolism**" throughout the manuscript. The locations and reasons for this revision are described below.

- **Line 98-99:** In this study, a comprehensive dataset encompassing material metabolism.
- **Material Metabolism:** Analogizing industrial systems to biological metabolism, this approach examines the input, transformation, circulation, and output of macro-material categories (e.g., metals, fossil fuels, biomass) within human activities.

Scale: Focuses on systemic levels (e.g., cities, nations, industrial chains), emphasizing resource circularity and sustainable design.

Typical applications: Urban mining development, circular economy policy design.
- **Substance Metabolism:** Tracks the pathways and fate of specific chemical substances or elements (e.g., carbon, phosphorus, lead, microplastics) within environmental-economic systems.

Scale: Targets molecular/elemental levels, revealing environmental risks such as contaminant migration and bioaccumulation.

Typical applications: Heavy metal pollution control, emerging contaminants management.
- The concept of material metabolism originated in biological sciences. In 1857, Moleschott proposed in *The Cycle of Life* that the essence of life is "an exchange process of energy and matter." During the 1970s, ecologists extended metabolic theory to industrial systems, introducing the concept of "Industrial Metabolism" to analyze regional pollutant flows through material balance methods¹. In a seminal 1989 *Scientific American* paper, General Motors scientists Frosch and Gallopoulos first advanced the "Industrial Ecosystem" framework, advocating for closed-loop waste valorization chains modeled on natural ecosystems. As a core paradigm of Industrial Ecology, Material Metabolism analysis quantifies material flows

(Material Flow Analysis, MFA), stock accumulation, and associated environmental impacts to enable multi-scale system optimization. Typical papers include: ‘*Efficiency stagnation in global steel production urges joint supply- and demand-side mitigation efforts*², and ‘*Country-specific net-zero strategies of the pulp and paper industry*³.

Response References

1. Fischer-Kowalski, M. *Society’s Metabolism*. doi:10.1162/jiec.1998.2.1.61.
2. Wang, P. et al. *Efficiency stagnation in global steel production urges joint supply- and demand-side mitigation efforts*. *Nat Commun* **12**, 2066 (2021).
3. Dai, M. et al. *Country-specific net-zero strategies of the pulp and paper industry*. *Nature* **626**, 327–334 (2024).

[Issue 2] Line 31-32. Please rephrase the sentence. It reads as if plastic pollution is reduced by leakage compared to 2060 BAU.

[Response] Accepted. We appreciate your identification of this inaccuracy. Given the potentially misleading nature of the original statement, we have implemented corresponding revisions.

- **Line 31-32:** Relative to the 2060 BAU, the system change achieves an 80% reduction in plastic leakage and a 63% decrease in GHG emissions.

[Issue 3] Line 55-57 Please rephrase this sentence. I can’t understand what the authors are trying to imply here.

[Response] Accepted. We appreciate your insightful suggestions. The methodological logic of this study is structured as follows: First, we establish high-resolution material flow data (covering 14 plastic types); then quantify environmental leakage and GHG emissions based on this foundation; finally conduct scenario analyses and related calculations. It is evident that the granular material flow data serve as the essential bedrock for subsequent research, both in terms of accounting accuracy and categorical refinement (14 types). Here, we emphasize the critical importance of refined material flow analysis to this study. we have implemented corresponding revisions.

- **Line 53-55:** The granularity of plastic Material Flow Analysis (MFA) fundamentally determines the accuracy in quantifying environmental leakage and GHG emissions, as it provides the foundational dataset for subsequent assessments.

[Issue 4] Line 67 What do you mean by low-volume plastic types? Is it based on the production volume or waste generation volume?

[Response] **Acknowledged.** We appreciate your insightful suggestions. The original wording obscured our intended meaning. Here we refer to lower-consumption plastic types such as PPS, PBT, and POM (accounting for merely 3.2% of total plastic consumption in 2021). Due to their limited consumption volume, these plastics receive insufficient research attention. Consequently, their use-phase lifespan and post-disposal fate are often overlooked in material flow accounting. Corresponding revisions have been implemented in the manuscript.

- **Line 66-68:** Particularly for low-consumption plastic types, the waste streams rely on empirical assumptions rather than field research data, further limiting the model's ability to generalize.

[Issue 5] Line 84-87 Why parallel to the study of GHG emissions. LCA can also quantify the GHG emissions in the form of Global Warming Potential/Carbon Footprint, can't they?

[Response] **Acknowledged.** We appreciate you highlighting this issue. The imprecise wording in the original version caused unintended ambiguity, for which we sincerely apologize. **This section separately reviews the research status of plastic environmental leakage and GHG emissions:** Life Cycle Assessment (LCA) constitutes the core methodological framework for GHG quantification, while Global Warming Potential (GWP) and carbon footprint represent two distinct forms of characterized emission outputs. To prevent misinterpretation, we have revised the content as follows.

- **Line 83-84:** For GHG emissions research related to plastics, the Life Cycle Assessment (LCA) methodology systematically analyzes the carbon emission distribution characteristics of the whole chain from fossil fuel extraction, polymer synthesis.....

[Issue 6] Line 91 - Unknown mechanisms for coupling environmental leakage. What mechanism did the author use to incorporate the environmental leakage data and results within the framework of your study?

[Response] **Acknowledged.** We appreciate your insightful inquiry. Elucidating the coupling mechanisms between environmental leakage and GHG emissions constitutes a key innovation of this study. Combining Eqs. 9-19 and Fig. 1 in the Supplementary Information, we analyzed the synergistic coupling mechanism between plastic leakage and GHG emissions through the following steps:

- **Systematically mapping emission pathways** for plastic leakage and associated GHG across the full life cycle via literature synthesis.
- **Categorizing the plastic life cycle into four key stages (Production, Transportation & Manufacturing, Use, Disposal)**, with 14 leakage pathways and 10 GHG emission pathways assigned to respective stages.
- **Integrating plastic MFA datasets, leakage factors, and GHG emission factors** to quantify leakage volumes and emissions per pathway/stage.
- **Comparing aggregate and stage-specific** emission profiles and historical trajectories.
- **Evaluating policy impacts through scenario modeling** to decipher dynamic coupling mechanisms by assessing co-variation of leakage and emissions under interventions. The ultimate objective is achieving **maximized synergistic reduction of leakage and emissions alongside minimized abatement costs** through optimized policy design.

Fig. 1 System boundaries and technical routes of MFA - Environmental leakage accounting - GHG emissions coupled model and scenario analysis.

● Supplementary Information – Eqs. 9-19

Plastic environmental leakage

$$Leakage_{total,k,t} = Leakage_{ma,k,t} + Leakage_{mj,k,t} \quad (9)$$

$Leakage_{total,k,t}$ refers to the total environmental leakage of type k plastics in year t, $Leakage_{ma,k,t}$ refers to the environmental leakage of macro-plastics of type k plastics in year t, $Leakage_{mj,k,t}$ refers to the environmental leakage of micro-plastics of type k plastics in year t.

$$Leakage_{ma,k,t} = Terr_{Ma,k,t} + Aqu_{Ma,k,t} \quad (10)$$

$Terr_{Ma,k,t}$ refers to the macroplastic leakage of type k plastics into the terrestrial environment in year t, $Aqu_{Ma,k,t}$ refers to the macroplastic leakage of type k plastics into the aquatic environment in year t.

$$Leakage_{mi,k,t} = Terr_{Mi,k,t} + Aqu_{Mi,k,t} \quad (11)$$

$Terr_{Mi,k,t}$ refers to the microplastic leakage of type k plastics to the terrestrial environment in year t, $Aqu_{Mi,k,t}$ refers to the microplastic leakage of type k plastics to the aquatic environment in year t.

$$Terr_{Ma,k,t} = Prod_{k,t} \times Prod_{Ma,\alpha} + Manuf_{k,t} \times Manuf_{Ma,\alpha} + Transp_{k,t} \times Transp_{Ma,\alpha} + Rec_{k,t} \times Rec_{Ma,\alpha} + Landfill_{k,t} \times Landfill_{Ma,\alpha} + Misman_{Ma,k,t} \quad (12)$$

$Prod_{k,t}$, $Manuf_{k,t}$, $Transp_{k,t}$, $Rec_{k,t}$ and $Landfill_{k,t}$ refer to the amount of plastic of type k that was produced, manufactured, transported, recycled and landfilled in year t, $Misman_{Ma,k,t}$ refers to the amount of plastic of type k that leaked into the land environment due to poor management of macroplastics in year t, $Prod_{Ma,\alpha}$, $Manuf_{Ma,\alpha}$, $Transp_{Ma,\alpha}$, $Rec_{Ma,\alpha}$ and $Landfill_{Ma,\alpha}$ refers to the leakage rate of plastics to the land environment in the stages of production, manufacture, transportation, recycling, and landfill. Leakage rate of macroplastics to the landfill.

$$Aqu_{Ma,k,t} = Aquac.ind_{k,t} + Fish.ind_{k,t} + River_{k,t} + Coastal_{k,t} \quad (13)$$

$Aquac.ind_{k,t}$, $Fish.ind_{k,t}$, $River_{k,t}$, $Rec_{k,t}$ and $Coastal_{k,t}$ refer to the amount of environmental leakage of type k plastics to the aquatic environment from aquaculture, fisheries, riverine areas, and coastal zones for the year t.

$$River_{k,t} = Misman_{k,t} \times \frac{Area_{land,river}}{Area_{land,total}} \times River_{ma,\alpha} \quad (14)$$

$Misman_{k,t}$ refers to the amount of mismanagement of type k plastics in year t, $Area_{land,river}$ refers to the total area covered by China's watersheds, $Area_{land,total}$ refers to the total area of China's national territory, $River_{ma,\alpha}$ refers to the rate of leakage of macroscopic plastics mismanagement from the river to the aquatic environment.

$$Coastal_{k,t} = \frac{Misman_{k,t}}{Pop_{land,total}} \times Area_{land,coastal} \times Coastal_{ma,\alpha} \quad (15)$$

$Pop_{land,total}$ refers to the total population of China in year t, $Area_{land,coastal}$ refers to the total coastal area of China, $Coastal_{ma,\alpha}$ refers to the leakage rate of poorly managed macroplastics from the coast to the aquatic environment.

$$Misman_{Ma,k,t} = Misman_{k,t} - River_{k,t} - Coastal_{k,t} \quad (16)$$

$$Terr_{Mi,k,t} = Transp_{k,t} \times Transp_{Mi,\alpha} + Indoor_{k,t} + Public_{k,t} + Landfill_{k,t} \times Landfill_{Mi,\alpha} \quad (17)$$

$Indoor_{k,t}$ and $Public_{k,t}$ refer to the amount of residential indoor dust, public indoor dust, and mismanagement of type k plastics in year t, $Transp_{Mi,\alpha}$ and $Landfill_{Mi,\alpha}$ refer to the leakage rate of microplastics into the terrestrial environment during the transport phase and landfill phase.

$$Aqu_{Mi,k,t} = Prod_{k,t} \times Prod_{Mi,\alpha} + Manuf_{k,t} \times Manuf_{Mi,\alpha} + PCP_{k,t} + laundry_{k,t} + Rec_{k,t} \times Rec_{Mi,\alpha} \quad (18)$$

$PCP_{k,t}$ and $laundry_{k,t}$ refer to the microplastic generation of type k plastics from personal-care supply and laundry in year t, $Prod_{Mi,\alpha}$, $Manuf_{Mi,\alpha}$ and $Rec_{Mi,\alpha}$ refer to the microplastic leakage rate to the aquatic environment from the production stage, the manufacturing stage and the recycling stage.

Plastics-related GHG emissions

$$GHG_{total,g,k,t} = ME_{g,k,t} + MP_{g,k,t} + Poly_{g,k,t} + PM_{g,k,t} + TM_{g,k,t} + Rec_{g,k,t} + Inc_{g,k,t} + Landfill_{g,k,t} - RS_{g,k,t} - IS_{g,k,t} \quad (19)$$

$GHG_{total,g,k,t}$ is the total amount of GHG emissions associated with type k plastics in year t, $ME_{g,k,t}$ is the GHG emissions associated with the extraction of feedstock for type k plastics in year t, $MP_{g,k,t}$ is the GHG emissions associated with the processing of a single plastic of type k in year t, $Poly_{g,k,t}$ is the polymerisation-related GHG emissions of type k plastics in year t, $PM_{g,k,t}$ is the primary processing-related GHG emissions of type k plastics in year t, $TM_{g,k,t}$ is the end-process-related GHG emissions of type k plastics in year t, $Rec_{g,k,t}$ is the GHG emissions associated with the recycling of type k plastics in year t, $Inc_{g,k,t}$ is the GHG emissions associated with the incineration of type k plastics in year t, $Landfill_{g,k,t}$ is the landfill-related GHG emissions of type k plastics in year t, $RS_{g,k,t}$ is the GHG emissions reductions associated with the recycling substitution of type k plastics in year t, $IS_{g,k,t}$ is the GHG emissions reductions from incineration substitution of type k plastics in year t.

[Issue 7] It will be ideal to concisely define the different EoL pathways considered in this manuscript so that the readers can understand the differences between EoL options like dumping, landfill or disposal.

[Response] Accepted. We appreciate your suggestion. We have supplemented the relevant content with supporting references in the Methods section.

Line 410-413: Current mainstream treatments for waste plastics comprise mechanical recycling, incineration, and landfilling. Chemical recycling has yet to achieve industry-scaled implementation but was incorporated into our scenario analysis^{19,52}. Open dumping and uncontrolled disposal are uniformly categorized as mismanagement^{2,16}.

Manuscript References

2. Cottom, J. W., Cook, E. & Velis, C. A. A local-to-global emissions inventory of macroplastic pollution. *Nature* **633**, 101–108 (2024).
16. Stokal, M. et al. River export of macro- and microplastics to seas by sources worldwide. *Nat. Commun.* **14**, 4842 (2023).
19. Lau, W. W. Y. et al. Evaluating scenarios toward zero plastic pollution. *Science* **369**, 1455–1461 (2020).
52. Chen, C., Wen, Z., Sheng, N. & Song, Q. Uneven agricultural contraction within fast-urbanizing urban agglomeration decreases the nitrogen use efficiency of crop production. *Nat. Food* **5**, 390–401 (2024).

Supplementary information - 3.7 Plastic waste disposal:

The definition and technical characteristics of the "end-of-life" (EoL) path for plastic waste in China are as follows:

1. Mechanical Recycling: Transforming single-type plastics (e.g., PET bottles, HDPE) back into recycled pellets or new products through physical processes like sorting, cleaning, shredding, and melt reprocessing. Relies on strict sorting and high cleanliness. Suitable for high-purity plastic waste.

2. Incineration: Co-incineration of plastics with municipal solid waste to generate electricity or heat using thermal energy. Offers significant volume reduction.

3. Landfilling: Co-disposal of plastics with other municipal solid waste in centralized sanitary landfills. Simple to operate but occupies land resources and offers no resource recovery.

4. Mismanagement / Dumping: Plastic waste discarded irregularly into the natural environment (land, water bodies, or illegal dumpsites), not integrated into formal management systems. Completely uncontrolled, directly causing ecological pollution.

5. Chemical Recycling: Converts plastics into fuel oil, carbon black, or combustible syngas. Can handle mixed/low-value plastics but is not yet commercially scalable.

[Issue 8] Line 124 What do you mean by metabolic map of plastics actually mean? Please rephrase or define the sentence.

[Response] Accepted. We appreciate you highlighting this issue. The plastic material metabolism map is a systematic research tool designed to analyze and visualize the flow, transformation, and

environmental fate of plastics throughout their full life cycle. This approach traces the dynamic migration of plastics across the supply chain—from raw material production and processing to end-of-life disposal. For a comprehensive definition and methodological details, please refer to the response to Question 1. We have revised this content accordingly.

- **Line 386-388:** We traced upstream to dissect the material flow through the chemical processes from fossil feedstocks to primary plastics, thereby refining the material metabolism of plastics (Supplementary Figure 2).

[Issue 9] Line 139 Primary plastic form as in polymers? Also please provide examples for what intermediates exactly mean.

[Response] Acknowledged. We appreciate your meticulous inquiry. Following Drewniok et al.'s⁴ paper, plastic flows are categorized into four stages: plastics in primary form, intermediates, end-use manufactured products, and waste.

- **Plastics in primary form:** Polymer-based raw materials (e.g., synthetic resins) formed through polymerization reactions, existing in unprocessed states such as pellets, powder, or liquid monomers.
- **Intermediates:** Semi-processed goods derived from primary plastic resins that require secondary manufacturing to become end-use products. Representative examples include:

Plastic sheets/plates: Formed by extrusion molding of PE/PP resins, ultimately used in food containers and stationery casings.

Plastic pipes: Produced via mold extrusion of PVC resins, primarily deployed as building water pipes and wire conduits.

Response References

4. Drewniok, M. P., Gao, Y., Cullen, J. M. & Cabrera Serrenho, A. *What to Do about Plastics? Lessons from a Study of United Kingdom Plastics Flows. Environ. Sci. Technol.* **57**, 4513–4521 (2023).

[Issue 10] Line 168 Estimation of plastic environmental leakage in China. How do you differentiate between macro and microplastics, and from which source you got the data to calculate such leakage values?

[Response] **Acknowledged.** We appreciate your meticulous inquiry. Leveraging high-resolution material flow analysis datasets and synthesizing literature-derived leakage rate parameters for macroplastic and microplastic pathways, we ultimately quantified life cycle-wide leakages of both macro- and microplastics. The validated leakage parameters and their bibliographic sources have been systematically enhanced in Supplementary Table 13 (Sources of Model Input Data). Detailed updates are specified below.

- **Supplementary Table 13. Sources of the data for model inputs.**

	Plastic Life Stages	Leakage direction	Leakage Sources	Leakage Parameters	Sources	
Macro-plastic	$Prod_{Ma,\alpha}$	Terrestrial	Total	1.000%	Luan, X. et al ⁹	
	$Manuf_{Ma,\alpha}$	Terrestrial	Total	0.750%	Luan, X. et al ⁹	
	$Transp_{Ma,\alpha}$	Terrestrial	Total	0.750%	Luan, X. et al ⁹	
	Use	$Aquac. ind_{k,t}$	Aquatic	-	-	Bai, M. et al ³⁷ , Luan, X. et al ⁹
		$Fish. ind_{k,t}$	Aquatic	-	-	Bai, M. et al ³⁷ , Luan, X. et al ⁹
	$Rec_{Ma,\alpha}$	Terrestrial	Recycling	0.400%	Luan, X. et al ⁹	
	$Landfill_{Ma,\alpha}$	Terrestrial	Landfill	0.100%	Luan, X. et al ⁹	
	$Coastal_{ma,\alpha}$	Aquatic	Mismanagement	1-5%	Strokal, M. et al ¹⁰ , Lau, W. W. Y. et al ³	
	$River_{ma,\alpha}$	Aquatic	Mismanagement	-	Strokal, M. et al ¹⁰	
$Misman_{Ma,k,t}$	Terrestrial	-	-	-		
Micro-plastic	$Prod_{Mi,\alpha}$	Aquatic	Total	0.050%	Luan, X. et al ⁹	
	$Manuf_{Mi,\alpha}$	Aquatic	Total	0.005%	Luan, X. et al ⁹	
	$Transp_{Mi,\alpha}$	Terrestrial	Total	0.002%	Luan, X. et al ⁹	
	Use	$PCP_{k,t}$	Aquatic	-	-	Strokal, M. et al ¹⁰ , Lau, W. W. Y. et al ³

	Laundry Products	Aquatic	Textiles consumption	0.120%	Strokal, M. et al ¹⁰ , Lau, W. W. Y. et al ³
	Indoor Dust	Terrestrial	-	0.080%	Strokal, M. et al ¹⁰ , Luan, X. et al ⁹
	Public Dust	Terrestrial	-	0.080%	Luan, X. et al ⁹
	$Rec_{Mi,\alpha}$	Aquatic	Recycling	0.030%	Lau, W. W. Y. et al ³ , Luan, X. et al ⁹
	$Landfill_{Mi,\alpha}$	Terrestrial	Landfill	0.120%	Lau, W. W. Y. et al ³ , Luan, X. et al ⁹

[Issue 11] For the IS_{kt} in the equation 7, what kind of substitution was considered for incineration? Is it just energy in general? Or electricity? Or electricity and heat? How much of is substituted in reality in China (BAU)?

[Response] Acknowledged. Regarding the substantial thermal energy generated during waste incineration, there are two primary pathways for its recovery and utilization: direct thermal use and Waste-to-Energy (WtE) power generation⁵. In China, direct thermal use is less commonly applied due to the typical distance limitations to suitable heat demand sites.

- WtE power generation is the predominantly utilized method. This process involves using superheated steam produced in the incineration boilers to drive turbine generators. The electricity generated not only powers the incineration facility itself but also creates economic benefits through feed-in to the power grid.
- More significantly, this electricity displaces power primarily derived from fossil fuels in the grid, thereby avoiding corresponding greenhouse gas (GHG) emissions⁶.
- Based on China's total incineration capacity, the calorific value proportion of plastics in waste streams, and their thermal conversion efficiency⁷, we calculated that plastic-derived power generation in 2021 displaced 23 Mt CO₂-eq of GHG emissions.

Response References

5. Chang Shoukui. *Comparative Study of High-Parameter Waste Incineration Power Generation Processes Based on Life Cycle Assessment*. (2014). (In Chinese)

6. Wang, L. L.; Zhang, X.; Zhang, Y.; Chen, Y. G.; Wan, J. F.; Wang, J.; Hu, L. M.. *Development and current status of domestic waste incineration power plants in China. Low Carbon World* 118-120 (2025) doi:10.16844/j.cnki.cn10-1007/tk.2025.03.054. (In Chinese)
7. Sang, Chunhui; Chen, Caozao; Wang, Po; Guo, Jianqiang; Hu, Ximin; Jia, Chong; Liu, Jing; Huang, Yue; Dong, Jingqi; Zhang, Hongzhen; Li, Xianglan. *Assessment and analysis of carbon emission reduction effect of domestic waste incineration power generation. Environmental Science* 2945-2952 (2025) doi:10.13227/j.hjcx.202405249. (In Chinese)

[Issue 12] Line 218 Please mention in the manuscript exactly what these different single and composite measures, even if you have mentioned them in the SI (section 7 and 8). Because, as a reader they need to know the specific measure for reducing the GHG emissions and leakage. Quantify the measures concisely in a table!!

[Response] Accepted. We appreciate your suggestion. Supplementing the relevant content will enhance the paper's comprehensiveness, and we have added this table to the scenario analysis section of the Methodology. It should be noted that when measures for a single scenario are modified, the relative proportions of other scenarios remain constant by default.

● **Line 506: Table 1 Specifications of 10 single scenarios and 4 composite scenarios.**

Scenario		By 2060
BAU		None
Single measure	Reform of the energy structure	1. Energy consumption in the power system: hydroelectricity (25%), coal (15%), nuclear (15%), wind (25%), photovoltaic (20%); Industrial steam production: coal (15%), natural gas (30%), solid waste (0.55%); Fuel consumption structure: coal (10%), natural gas (10%), solid waste (15%), primary energy and others (65%).
	Design optimization for reduction	2. The packaging sector targets 30% source reduction through design optimization, while prohibiting

		production of plastic-microbead household chemicals and PE mulch films below 0.01mm thickness.
Composite measures	Reduction in demand	3. Demand in the packaging industry decreased by 30%.
	Reduction through substitution	4. Five packaging substitutes (bamboo, wood, paper, glass, biodegradable plastics) equally share a 30% market substitution.
	Collection	5. Mismanaged waste incidence reduced to 1%.
	Mechanical recycling	6. Mechanical recycling achieves 50%.
	Chemical recycling	7. Chemical recycling achieves 20%.
	Incineration	8. Incineration achieves 60%.
	Landfilling	9. Landfilling achieves 30%.
	Reduction and substitution (1-4)	10. Reform of the energy structure. The packaging sector targets 90% source reduction, while prohibiting production of plastic-microbead household chemicals and PE mulch films below 0.01mm thickness.
	Recycling (6-7)	11. Mechanical recycling achieves 35%. Chemical recycling achieves 25%.
	Collect and dispose (5, 8-9)	12. Mismanaged waste incidence reduced to 1%. Incineration achieves 50%. Landfilling achieves 19%.
	System change (1-9)	13. Reform of the energy structure. The packaging sector targets 90% source reduction, while prohibiting production of plastic-microbead household chemicals and PE mulch films below 0.01mm thickness. Mismanaged waste incidence reduced to 1%. Mechanical recycling achieves 30%. Chemical recycling achieves 18%.

Incineration achieves 30%.

Landfilling achieves 21%.

[Issue 13] Line 232-233 Recycling and incineration can generate revenue is a generic statement. Either explain it a bit or quantify them citing the relevant source.

[Response] **Acknowledged.** We appreciate your comment. We have provided further clarification on this content in the relevant sections and supplemented the literature citations accordingly.

- **Line 515-518:** Recycled plastics are processed into regenerated pellets that substitute virgin plastic feedstocks, thereby reducing upstream production costs in the industrial chain^{19,61}. Incineration generates direct economic revenue through energy recovery (electricity/heat generation)^{42,62}.

Manuscript References

19.Lau, W. W. Y. *et al.* Evaluating scenarios toward zero plastic pollution. *Science* **369**, 1455–1461 (2020).

42.Pottinger, A. S. *et al.* Pathways to reduce global plastic waste mismanagement and greenhouse gas emissions by 2050. *Science* **386**, 1168–1173 (2024).

61.Kumar, M. *et al.* Retrieving back plastic wastes for conversion to value added petrochemicals: opportunities, challenges and outlooks. *Appl. Energy* **345**, 121307 (2023).

62.Lim, M., Lee, Y., Lee, Y., Yang, W. & Kim, S. Improving waste-incineration energy recovery efficiency using a reverse calculation algorithm to estimate waste composition and heating value. *Waste Manag.* **190**, 486–495 (2024).

[Issue 14] Line 236 'by removing the revenues..' why?

[Response] **Accepted.** We appreciate you highlighting this issue. Our initial ambiguous phrasing failed to convey the intended meaning: **subtracting plastic-related revenues from waste management expenditures yields the net economic cost of plastic waste management in China under various future scenarios.** Plastic-related revenues primarily stem from recycled material substitution for virgin plastic feedstocks and energy recovery via incineration. Expenditures and revenues associated with plastic pollution control constitute pivotal factors in projecting net

economic costs for synergistic plastic governance in China. To forecast the costs of different measures from 2022 to 2060, this study systematically compiled unit cost and unit revenue data for plastic waste management activities. We have revised it for enhanced clarity.

- **Line 519-522:** the net economic costs of plastic waste management in China can be projected for different future scenarios from 2022 to 2060 by subtracting plastic-related revenues from waste management cost expenditures.

[Issue 15] Line 245 1000 Monte Carlo simulations. What parameters or what kind of data were run in Monte-Carlo analysis and why? Due to the uncertainty of data?

[Response] Acknowledged. In material flow analysis, Monte Carlo analysis is an effective tool for assessing the impact of uncertainty in model input parameters on the model's predictive outcomes. In plastic material flow analysis, the uncertainty range for each input variable is determined based on the quality and source of the data (i.e., the data pedigree). These uncertainties naturally propagate throughout the Monte Carlo simulation process, thereby affecting the predicted outcomes of plastic flow and quality changes. To obtain more precise predictions, we conducted 1000 Monte Carlo simulations of the plastic flow process. The relevant value-taking procedure is in Section 10 of the supplementary material.

[Issue 16] Line 253, 255, 260 - Please cite the websites appropriately.

[Response] Accepted. We appreciate your suggestion. Following conventions in peer-reviewed studies, we have standardized the citation format for web-based references.

- **Line 416-425:** (2) Trade data on primary plastic products, finished plastic products and plastic waste can be obtained from the online statistical data service platform of the General Administration of Customs of the People's Republic of China⁵⁴ and the United Nations Comtrade database⁵⁵; (3) Parameters for environmental leakage and GHG emissions were derived from the literature, with primary sources detailed in the Supplementary Information sections 4 and 5; (4) Population estimates can be obtained from the "Forecast of Medium and Long Term Change Trend of Chinese Population" compiled by the China Population and Development Research Center and the United Nations Population Fund China Representative Office⁵⁶;

Manuscript References

54. United Nations. UN Comtrade (2025). <https://comtradeplus.un.org/>.

55. General Administration of Customs People's Republic of China. Customs statistics (2025). <http://stats.customs.gov.cn/>.

56. China's Population Projection-Medium Variant (2021-2050). *UNFPA China* <https://china.unfpa.org/zh-Hans/publications/22070101>.

[Issue 17] Figure 2 need to be redone completely. Colour codes are confusing. Similar colours for incineration, landfill in the S4. Disposal. How do you differentiate between landfill and dumping? Because sometime landfill can be uncontrolled. What do you mean by Terminal manufacture? Why collection and sorting is not a part in Disposal? Please clarify.

[Response] Accepted. We appreciate your suggestion. The following revisions have been implemented:

- Fig. 2: Split into main Fig. 2 and Supplementary Figure 7, with X-axis labels supplemented.

- **How to distinguish landfilling from dumping?**

Answer: China classifies "uncontrolled landfills" lacking impermeable liner systems as dumping. We quantified plastic landfilling in China by: (a) Calculating the total mass of sanitary landfills that meet national standards. (b) Extrapolating plastic proportion in landfilled waste. (c) Cross-validating results with China's disposal ratios from Luan et al⁸.

- **What do you mean by Terminal manufacture?**

Final Manufacturing: The process of converting plastic feedstocks (virgin resin/recycled pellets) into end-use products ready for direct consumption, distinct from raw material production (e.g., petrochemical refining) and intermediate processing. Intermediate Processing: Physicochemical treatments that transform base plastic materials into standardized intermediates (e.g., pipes, sheets), where outputs require further manufacturing to become final products. Example: PE pellets extruded into pipes must be cut and welded before functioning as water conduits.

- **Why collection and sorting is not a part in Disposal?**

We appreciate this insightful inquiry. Collection and sorting constitute critical stages in plastic recycling, where their efficiency and costs directly constrain overall recovery rates. In our

historical pathway analysis (1992–2021), we adopted a result-oriented principle: (a) The impacts of collection/sorting are embedded in observable recycling rate variations (derivable from public data), while (b) Disposal metrics (e.g., incineration rates) are independently accessible, thus eliminating the need for separate collection pathway modeling. However, in scenario analysis, we designed a single policy scenario: upgrading recycling rates, and a combination scenario: upgrading both recycling and sorting technologies to quantify the impact of future technologies on plastic leakage, greenhouse gas emissions, and cost-effectiveness dynamics.

Response References

8. Luan, X. *et al.* *Dynamic material flow analysis of plastics in China from 1950 to 2050. Journal of Cleaner Production* **327**, 129492 (2021).

Fig. 2 Environmental leakage and associated GHG emissions from plastics in China, 1992-2021. **a**, Annual environmental leakage from 14 pathways in four stages of plastics in China, 1992-2021. **b**, Annual GHG emissions from 10 pathways in four stages of plastics in China, 1992-2021. The black line indicates net GHG emissions, i.e., total GHG emissions minus total GHG offsets. Offsets include the substitution of raw materials for virgin plastics production by recovered recycled plastics and the substitution of carbon emissions from fossil energy electricity by plastics incineration for electricity generation. **c**, Cumulative environmental leakage of plastics and associated GHG emissions in China from 1992-2021, divided into four stages: Stage 1 (production),

Stage 2 (transport and manufacturing), Stage 3 (use) and Stage 4 (disposal). Corresponds to the partition in the illustration. **d**, Cumulative emissions from environmental leakage of plastics in China from 1992-2021, divided into 14 pathways. **e**, Cumulative plastics-related net GHG emissions in China from 1992 to 2021, divided into 10 pathways.

Supplementary Figure 7. Total macroplastic and microplastic leakage of plastics and their associated GHG emissions in China from 1992 to 2021. **a**, Microplastic leakage. **b**, Macroplastic leakage. **c**, Associated GHG emissions.

[Issue 18] Line 292 Long-life products with textiles. Is it really the case? Any source to substantiate these claims?

[Response] Acknowledged. We appreciate you highlighting this inaccuracy. The original statement lacked precision. By the mid-1990s, synthetic fiber production surpassed cotton, growing substantially from 10 million tonnes (Mt) in 1975 to 87.6 Mt in 2022⁹. Although plastic fibers in

textiles typically have service lives under 10 years^{8,10,11}, their massive consumption volume results in significant in-use stocks. We have revised this content accordingly.

- **Line 195-198:** Plastic stocks increased from 8.0 Mt in 1992 to 487.0 Mt in 2021, which is 4.6 times the amount of plastic consumed. Long-life products dominate the plastics stock¹¹. From an industry perspective, construction and textiles accounting for 64.4% of the stock (Fig. 3b).

References

8. Luan, X. et al. *Dynamic material flow analysis of plastics in China from 1950 to 2050. Journal of Cleaner Production* 327, 129492 (2021).
9. Ning, J. et al. *Substance flow analysis of polyethylene terephthalate (PET) fiber in China. Resources, Conservation and Recycling* 212, 107984 (2025).
10. Chu, J. et al. *Dynamic flows of polyethylene terephthalate (PET) plastic in China. Waste Management* 124, 273–282 (2021).
11. Geyer, R., Jambeck, J. R. & Law, K. L. *Production, use, and fate of all plastics ever made. Science Advances* 3, e1700782 (2017).

[Issue 19] If you could define/quantify the measures for different scenarios you can discuss them better in the Discussion section.

[Response] Accepted. We appreciate your suggestion. Supplementing the relevant content will enhance the paper's comprehensiveness, and we have added this table to the scenario analysis section of the Methodology. It should be noted that when measures for a single scenario are modified, the relative proportions of other scenarios remain constant by default.

- **Line 506:** Table 1 Specifications of 10 single scenarios and 4 composite scenarios.

Scenario		By 2060
BAU		None
Single measure	Reform of the energy structure	1. Energy consumption in the power system: hydroelectricity (25%), coal (15%), nuclear (15%), wind (25%), photovoltaic (20%); Industrial steam production: coal (15%), natural gas (30%), solid waste (0.55%);

		Fuel consumption structure: coal (10%), natural gas (10%), solid waste (15%), primary energy and others (65%).
	Design optimization for reduction	2. The packaging sector targets 30% source reduction through design optimization, while prohibiting production of plastic-microbead household chemicals and PE mulch films below 0.01mm thickness.
	Reduction in demand	3. Demand in the packaging industry decreased by 30%.
	Reduction through substitution	4. Five packaging substitutes (bamboo, wood, paper, glass, biodegradable plastics) equally share a 30% market substitution.
	Collection	5. Mismanaged waste incidence reduced to 1%.
	Mechanical recycling	6. Mechanical recycling achieves 50%.
	Chemical recycling	7. Chemical recycling achieves 20%.
	Incineration	8. Incineration achieves 60%.
	Landfilling	9. Landfilling achieves 30%.
Composite measures	Reduction and substitution (1-4)	10. Reform of the energy structure. The packaging sector targets 90% source reduction, while prohibiting production of plastic-microbead household chemicals and PE mulch films below 0.01mm thickness.
	Recycling (6-7)	11. Mechanical recycling achieves 35%. Chemical recycling achieves 25%.
	Collect and dispose (5, 8-9)	12. Mismanaged waste incidence reduced to 1%. Incineration achieves 50%. Landfilling achieves 19%.
	System change (1-9)	13. Reform of the energy structure. The packaging sector targets 90% source reduction, while prohibiting production of plastic-microbead

household chemicals and PE mulch films below 0.01mm thickness.

Mismanaged waste incidence reduced to 1%.

Mechanical recycling achieves 30%.

Chemical recycling achieves 18%.

Incineration achieves 30%.

Landfilling achieves 21%.

[Issue 20] I didnt understand Fig 6. Why there is a separate graph for different polymer types?

[Response] **Acknowledged.** We appreciate this profound insight.

- A core innovation of this study lies in its comprehensive and fine-grained multi-dimensional analysis of 14 plastics, systematically integrating five key modules: material flows, environmental leakage, GHG emissions, scenario projections, and cost-benefit assessments. We hope to be able to analyze the differences in flow, environmental impact, and pollution control between the 14 plastic types.
- In the cost-benefit analysis, we quantified the differences in economic costs for all 14 scenarios.
- And the combined analysis shows that the system change scenario achieves the optimal synergy between emission reduction and cost-effectiveness.
- Therefore, in conjunction with the visualization constraints, we would like to label the cost-effectiveness metrics of the 14 plastics under the system change scenarios in the figure.

Reviewer # 3

[Issue 1] Line 40: fourth in the world in: stats here would be interesting.

[Response] Acknowledged. Thanks for your careful comment. In country-level studies, Jambeck et al.¹² quantified the mismanaged plastic waste volumes of nations in 2010 through their seminal 2015 paper "Plastic waste inputs from land into the ocean" published in Science. According to the study, China ranked first globally with mismanaged plastic waste reaching 8.82 million tons per year (Mt/yr), significantly exceeding other countries. These findings have sparked extensive attention and debate within the academic community, with a portion of scholars suggesting that their estimates may be overstated^{13–15}. In their 2024 study published in Nature, titled "A local-to-global emissions inventory of macroplastic pollution", Cottom et al.¹⁶ employed more rigorous data and accounting methods to quantify global macroplastic pollution emissions for 2020. The results indicate: *"On an absolute basis, we find that plastic pollution emissions are highest across countries in Southern Asia, Sub-Saharan Africa and South-eastern Asia, with the largest amount (9.3 Mt year⁻¹ [6.5–12.7]) emitted by India, equivalent to nearly one-fifth of global plastic emissions. In contrast to previous plastic pollution models that positioned China as the world's highest plastic polluter, it is ranked fourth in our results, with emissions of 2.8 Mt year⁻¹ [2.1–3.7], less than Nigeria (3.5 Mt year⁻¹ [2.6–4.6]) and Indonesia (3.4 Mt year⁻¹ [2.5–4.3]). This lower contribution to plastic emissions from China reflects our use of more up-to-date data³⁵ that shows its substantial progress in adopting waste incineration and controlled landfill."*

References

12. Jambeck, J. R. et al. Plastic waste inputs from land into the ocean. *Science* **347**, 768–771 (2015).
13. Stokal, M. et al. River export of macro- and microplastics to seas by sources worldwide. *Nat Commun* **14**, 4842 (2023).
14. Mai, L. et al. Global Riverine Plastic Outflows. *Environ. Sci. Technol.* **54**, 10049–10056 (2020).
15. Mai, L., Sun, X. & Zeng, E. Y. Country-specific riverine contributions to marine plastic pollution. *Science of The Total Environment* **874**, 162552 (2023).
16. Cottom, J. W., Cook, E. & Velis, C. A. A local-to-global emissions inventory of macroplastic pollution. *Nature* **633**, 101–108 (2024).

[Issue 2] Line 43: are linked to China⁵: Is this because the plastics are produced in China or because they are used in China? Some of both?

[Response] Acknowledged. Thanks for your careful comment. We posit that such outcomes stem from the synergistic effect of both factors. The evidence is detailed below:

- In terms of the global pattern of plastic emissions, *since 1995, China's plastics-related carbon footprint has more than tripled from both a production and a consumption perspective. In 2015, 40% of the global plastics-related carbon footprint and more than 60% of the related coal-based emissions were caused in China. Almost half of the plastics produced in China and South Africa were exported, such as to the European Union and to the United States*¹⁷. The results show that **the growth in demand for plastics produced in coal-based economies, such as China in particular, has contributed significantly to the increase in the global carbon footprint of plastics.**
- Analysis of phase-specific emission profiles reveals that per tonne of plastic produced, carbon emissions are distributed as follows across the lifecycle: approximately 0.92 t CO₂-eq from raw material extraction, 1.66 t CO₂-eq from monomer cracking, 0.69 t CO₂-eq from polymerization, 0.94 t CO₂-eq from manufacturing, 0.27 t CO₂-eq from recycling processes (offset by 2.40 t CO₂-eq through recycled plastics utilization), 0.08 t CO₂-eq from landfilling, and 2.33 t CO₂-eq from incineration (offset by 0.90 t CO₂-eq via energy recovery).
- Classified into four core phases—production, manufacturing, use, and disposal—China generated 170 million tonnes (Mt) of plastics in 2021, with 80 Mt entering disposal (recycling: 24 Mt; incineration: 26 Mt; landfilling: 31 Mt). Total plastics-related GHG emissions reached 760 Mt, yielding net emissions of 679 Mt after accounting for 81 Mt of offsets from recycling and incineration energy recovery. **The production-manufacturing phases contributed 691 Mt (91% of total emissions), while the disposal phase emitted 69 Mt (incineration: 60 Mt, 8%).**
- **Overall, this emission pattern primarily stems from China's pivotal role in global plastic production chains. Crucially, rapid expansion of waste-to-energy infrastructure drove the plastic waste incineration rate from 3% in 2000 to 33% in 2020. Although disposal currently represents a smaller emission share, its growth rate significantly outpaces other lifecycle phases.**

References

17. Cabernard, L., Pfister, S., Oberschelp, C. & Hellweg, S. Growing environmental footprint of plastics driven by coal combustion. *Nat Sustain* **5**, 139–148 (2022).

[Issue 3] Line 72: synergistic assessment of environmental leakage does anyone ever try to directly measure environmental leakage, or it is just inferred by saying the difference between (production+ imports) and (landfill + recycling) = leakage? or something similar to this?

[Response] **Acknowledged.** Thanks for your careful comment. In fact, Within the field of environmental plastic leakage research, both direct measurement (bottom-up) and theoretical accounting (top-down) methodologies have been developed, yet they exhibit significant differences in their application scenarios and research focus. Given the diverse pathways of plastic leakage into the environment, direct measurement techniques are primarily applied to point source monitoring, such as sea shoreline¹⁸, rivers¹⁴, Coastal¹⁹, and wastewater treatment plants²⁰. In contrast, theoretical accounting is typically employed for regional-scale emissions assessments^{12,16,21}. At this scale, comprehensive direct measurement becomes impractical due to the complexity of all leakage pathways, necessitating reliance on theoretical approaches. Underpinned by a top-down research framework, this study refines emission accounting precision by systematically delineating various leakage pathways and integrating parameters derived from direct measurements reported in the literature.

Response References

12. Jambeck, J. R. et al. Plastic waste inputs from land into the ocean. *Science* **347**, 768–771 (2015).
14. Mai, L. et al. Global Riverine Plastic Outflows. *Environ. Sci. Technol.* **54**, 10049–10056 (2020).
16. Cottom, J. W., Cook, E. & Velis, C. A. A local-to-global emissions inventory of macroplastic pollution. *Nature* **633**, 101–108 (2024).
18. Terzi, Y. et al. Marine litter on the Turkish Black Sea shoreline: Abundance, composition, and sources. *Waste Management* **205**, 115027 (2025).
19. Kaandorp, M. L. A., Lobelle, D., Kehl, C., Dijkstra, H. A. & van Sebille, E. Global mass of buoyant marine plastics dominated by large long-lived debris. *Nat. Geosci.* **16**, 689–694 (2023).

20. Murphy, F., Ewins, C., Carbonnier, F. & Quinn, B. Wastewater Treatment Works (WwTW) as a Source of Microplastics in the Aquatic Environment. *Environ. Sci. Technol.* **50**, 5800–5808 (2016).

21. Luan, X. et al. Estimation and prediction of plastic losses to the environment in China from 1950 to 2050. *Resources, Conservation and Recycling* **184**, 106386 (2022).

[Issue 4] Line 77: non-end: non-end = ? I am not familiar with this term.

[Response] **Acknowledged.** Thank you for pointing this out. Given the contextual ambiguity in the initial phrasing, this definition has been refined in the full manuscript.

- **Line 76-78:** In addition, there is a lack of systematic quantification of plastics leakage in production and manufacturing processes, such as fugitive trimmings and microplastics by-products.

[Issue 5] Line 91: Unknown: lowercase.

[Response] **Accepted.** We appreciate your identification of this inconsistency and have refined the relevant description in the manuscript as prompted.

- **Line 90:** unknown mechanisms for coupling environmental leakage and GHG emissions, and a lag in updating non-mainstream plastics databases.

[Issue 6] Line 223: For more details, please refer to sections 7: I feel like you need to explain what these Scenarios are in the main text, this is very important to understanding what you are trying to do in this paper.

[Response] **Accepted.** We appreciate your suggestion. Supplementing the relevant content will enhance the paper's comprehensiveness, and we have added this table to the scenario analysis section of the Methodology. It should be noted that when measures for a single scenario are modified, the relative proportions of other scenarios remain constant by default.

- **Line 506:** Table 1 Specifications of 10 single scenarios and 4 composite scenarios.

Scenario	By 2060
BAU	None

Single measure	Reform of the energy structure	1. Energy consumption in the power system: hydroelectricity (25%), coal (15%), nuclear (15%), wind (25%), photovoltaic (20%); Industrial steam production: coal (15%), natural gas (30%), solid waste (0.55%); Fuel consumption structure: coal (10%), natural gas (10%), solid waste (15%), primary energy and others (65%).
	Design optimization for reduction	2. The packaging sector targets 30% source reduction through design optimization, while prohibiting production of plastic-microbead household chemicals and PE mulch films below 0.01mm thickness.
	Reduction in demand Reduction through substitution	3. Demand in the packaging industry decreased by 30%. 4. Five packaging substitutes (bamboo, wood, paper, glass, biodegradable plastics) equally share a 30% market substitution.
	Collection	5. Mismanaged waste incidence reduced to 1%.
	Mechanical recycling	6. Mechanical recycling achieves 50%.
	Chemical recycling	7. Chemical recycling achieves 20%.
	Incineration	8. Incineration achieves 60%.
	Landfilling	9. Landfilling achieves 30%.
Composite measures	Reduction and substitution (1-4)	10. Reform of the energy structure. The packaging sector targets 90% source reduction, while prohibiting production of plastic-microbead household chemicals and PE mulch films below 0.01mm thickness.
	Recycling (6-7)	11. Mechanical recycling achieves 35%. Chemical recycling achieves 25%.
	Collect and dispose	12. Mismanaged waste incidence reduced to 1%.

(5, 8-9)	Incineration achieves 50%. Landfilling achieves 19%.
System change (1-9)	13. Reform of the energy structure. The packaging sector targets 90% source reduction, while prohibiting production of plastic-microbead household chemicals and PE mulch films below 0.01mm thickness. Mismanaged waste incidence reduced to 1%. Mechanical recycling achieves 30%. Chemical recycling achieves 18%. Incineration achieves 30%. Landfilling achieves 21%.

[Issue 7] Line 236: revenues from plastic waste management expenditures: does this mean the revenues associated with collected plastics and selling them as a secondary recycled feedstock?

[Response] **Acknowledged.** We appreciate you highlighting this issue. Our initial ambiguous phrasing failed to convey the intended meaning: **subtracting plastic-related revenues from waste management expenditures yields the net economic cost of plastic waste management in China under various future scenarios.** Plastic-related revenues primarily stem from recycled material substitution for virgin plastic feedstocks and energy recovery via incineration. Expenditures and revenues associated with plastic pollution control constitute pivotal factors in projecting net economic costs for synergistic plastic governance in China. To forecast the costs of different measures from 2022 to 2060, this study systematically compiled unit cost and unit revenue data for plastic waste management activities. We have revised it for enhanced clarity.

- **Line 519-522:** the net economic costs of plastic waste management in China can be projected for different future scenarios from 2022 to 2060 by subtracting plastic-related revenues from waste management cost expenditures.

[Issue 8] Line 252-254: references are switched.

[Response] Accepted. We appreciate your suggestion. Following conventions in peer-reviewed studies, we have standardized the citation format for web-based references.

- **Line 416-425:** (2) Trade data on primary plastic products, finished plastic products and plastic waste can be obtained from the online statistical data service platform of the General Administration of Customs of the People's Republic of China⁵⁴ and the United Nations Comtrade database⁵⁵; (3) Parameters for environmental leakage and GHG emissions were derived from the literature, with primary sources detailed in the Supplementary Information sections 4 and 5; (4) Population estimates can be obtained from the "Forecast of Medium and Long Term Change Trend of Chinese Population" compiled by the China Population and Development Research Center and the United Nations Population Fund China Representative Office⁵⁶;

Manuscript References

54. United Nations. UN Comtrade (2025). <https://comtradeplus.un.org/>.

55. General Administration of Customs People's Republic of China. Customs statistics (2025). <http://stats.customs.gov.cn/>.

56. China's Population Projection-Medium Variant (2021-2050). *UNFPA China* <https://china.unfpa.org/zh-Hans/publications/22070101>.

[Issue 9] Line 270: Fig.2. Environmental leakage and associated: this is far too busy for a single Figure, in my opinion. I think a few of these Figures should move to be their own Figure, or enlarged by making this a full-page Figure, because right now they aren't really legible.

[Response] Accepted. We appreciate your suggestion. The following revisions have been implemented:

- Fig. 2: Split into main Fig. 2 and Supplementary Figure 7, with X-axis labels supplemented.

Fig. 2 Environmental leakage and associated GHG emissions from plastics in China, 1992-2021. **a**, Annual environmental leakage from 14 pathways in four stages of plastics in China, 1992-2021. **b**, Annual GHG emissions from 10 pathways in four stages of plastics in China, 1992-2021. The black line indicates net GHG emissions, i.e., total GHG emissions minus total GHG offsets. Offsets include the substitution of raw materials for virgin plastics production by recovered recycled plastics and the substitution of carbon emissions from fossil energy electricity by plastics incineration for electricity generation. **c**, Cumulative environmental leakage of plastics and associated GHG emissions in China from 1992-2021, divided into four stages: Stage 1 (production),

Stage 2 (transport and manufacturing), Stage 3 (use) and Stage 4 (disposal). Corresponds to the partition in the illustration. **d**, Cumulative emissions from environmental leakage of plastics in China from 1992–2021, divided into 14 pathways. **e**, Cumulative plastics-related net GHG emissions in China from 1992 to 2021, divided into 10 pathways.

Supplementary Figure 7. Total macroplastic and microplastic leakage of plastics and their associated GHG emissions in China from 1992 to 2021. **a**, Microplastic leakage. **b**, Macroplastic leakage. **c**, Associated GHG emissions.

[Issue 10] Line 284: And the cumulative emissions and offset: this needs more explanation.

[Response] Accepted. We thank you for this insight. As annotated in Fig. 2:

- Cumulative emissions: Total GHG emissions from plastics (1992–2021).
- Cumulative offsets: Sum of (a) material offsets from plastic recycling replacing virgin plastic production, and (b) energy offsets from waste-to-electricity displacing fossil-based power generation.

[Issue 11] Line 295: then slowly decreased to 6.9 Mt (Ter. 6.4, Aq. 0.5) in 2016: what accounted for this decrease in leakage? new policies, etc?

[Response] **Acknowledged.** We appreciate your inquiry. As noted, this question has been addressed in subsequent sections. Prompted by editorial requirements, the manuscript has been restructured with relevant content now located on **line 153-160**.

[Issue 12] Line 297: emissions increased from 21.9 Mt CO₂-eq (Emiss. 22.5, Offset. 0.6) in 1992 to 679.1 Mt CO₂-eq: do the emissions increase more or less than the overall plastics production? why?

[Response] **Acknowledged.** We appreciate your inquiry.

- From 1992 to 2021, **China's primary plastics production** surged from 4.7 Mt to 166.4 Mt, at a compound annual growth rate of **12.7%**.
- Concurrently, the annual **net GHG emissions** increased from 21.9 Mt CO₂-eq (Emiss. 22.5, Offset. 0.6) in 1992 to 679.1 Mt CO₂-eq (Emiss. 759.9, Offset. 80.8) in 2021, representing a compound annual growth rate of **12.1%**. The total emissions growth rate throughout the entire life cycle of plastics is 12.5% (22.5 to 759.9), while the emission reductions achieved through plastics recycling and waste-to-energy incineration grew at a rate of 9.1% (0.6 to 80.8).
- **Overall analysis indicates that** the growth rate of plastics production slightly outpaced that of its net emissions during this period (1992-2021). The primary reason for this disparity is the effective reduction in GHG emission intensity per unit of production, achieved through technological upgrades and process improvements in plastics manufacturing.

[Issue 13] Line 315-p319: here we go, this answers my previous question.

[Response] **Acknowledged.** Thank you for your question.

[Issue 14] Line 338: lowercase.

[Response] **Accepted.** We appreciate your identification of this inconsistency and have refined the relevant description in the manuscript as prompted. As requested by the editors, the manuscript has been restructured, with revisions implemented at the locations specified below.

- **Line 177:** processes of plastics in China in 2021.

[Issue 15] Line 339: Figure 3a question - what are the inputs and outputs of the digram on the far righthand side of Figure 3a, why don't they match the other numbers I see in Figure 3a? more explanation in the Figure caption is needed.

[Response] Accepted. We appreciate you highlighting this issue. We acknowledge insufficient interpretation of this figure segment. The right panel of Fig. 3a (Fig. 3b) illustrates in-use stocks of plastic polymer types across industrial sectors. This contrasts with the left panel's representation of instantaneous plastic material flows in 2021, as the right panel captures cumulative stock dynamics, inherently differing in data magnitude.

Fig. 3 Analysis of plastics material flows, environmental leakage, and GHG emissions in China in 2021. **a**, Plastics from fossil material to treatment and disposal in China in 2021. **b**, In-use stocks of 14 types of plastics in 10 sectors, 1992-2021. **c**, Environmental leakage accounting for 4 stages and 14 pathways for plastics in China in 2021. **d**, GHG emissions accounting for 4 stages and 10 processes of plastics in China in 2021.

[Issue 16] Line 431: from 73% to 91% and net GHG reduction impacts: how do you get these percentages from Fig 5?

[Response] **Acknowledged.** We appreciate your identification of this matter. This ratio is derived from a comparative analysis between the System Change scenario and the Business-as-Usual (BAU) scenario. Given significant variations in mitigation potential across plastic types, this study quantifies emission reduction effects for all plastic categories under each scenario, with focused comparison on these two specific scenarios. Environmental leakage and GHG emissions data for the 14 plastic types under the 2060 BAU scenario are detailed in Supplementary Section 8.1 (BAU Scenario). Additionally, figure references have been incorporated herein. As requested by the editors, the manuscript has been restructured, with revisions implemented at the locations specified below.

- **Line 276:** (Fig. 5, Supplementary Figure 12).

[Issue 17] Line 448: how are you estimating your chemical recycling costs, these are pretty immature technologies still?

[Response] **Acknowledged.** We appreciate your meticulous inquiry regarding this matter. For data on the costs and benefits of plastic chemical recycling, we referenced the 2020 study **Evaluating Scenarios Toward Zero Plastic Pollution** published in *Science* by Lau et al.²² Detailed documentation of chemical recycling costs and post-recycling material selling prices can be found in Sections S16.7 and S16.8 of the study's supplementary materials.

Response References

22. Lau, W. W. Y. et al. *Evaluating scenarios toward zero plastic pollution. Science* **369**, 1455–1461 (2020).

[Issue 18] Supplementary information - Line 439: table heading needs to be with Table?

[Response] **Accepted.** We sincerely thank you for this important formatting suggestion. We are grateful to the reviewers for pointing out this omission. The heading of Supplementary Table 13 of the Supplementary Information has now been repositioned on the same page as the table to conform to the required format.

[Issue 19] Supplementary information - Line 769: should be a brief text explanation of what's going on here?

[Response] Accepted. We sincerely thank the reviewers for their insightful suggestion, which prompted us to add explanatory text to better emphasize to the reader the strategic focus of this scenario.

- **Supplemental Information, below Section 8.11 (Reductions and Substitutions):** This integrated scenario combines reform of the energy structure (Section 8.2), design optimization for reduction (Section 8.3), reduction in demand (Section 8.4), and reduction through substitution (Section 8.5). This scenario focuses on the synergistic emission reduction effects of source reduction and energy transformation at the front end of the supply chain. Under baseline assumptions maintaining constant scenario weightings during individual variable modifications, key targets include: (1) Reform of the energy structure. (2) 90% source reduction in packaging through design optimization alongside prohibitions on plastic-microbead household chemicals and sub-0.01mm PE mulch films.

[Issue 20] Supplementary information - Line 774: brief text to explain?

[Response] Accepted. We sincerely thank the reviewers for their insightful suggestion, which prompted us to add explanatory text to better emphasize to the reader the strategic focus of this scenario.

- **Supplemental Information, below Section 8.12 (Recycling):** This scenario integrates mechanical recycling (Section 8.7) and chemical recycling (Section 8.8), with a particular emphasis on the synergistic emission reduction benefits that can be achieved by increased recycling rate strategies alone. Under baseline assumptions maintaining constant scenario weightings during individual variable modifications, key targets include: (1) mechanical recycling achieving 35%; and (2) chemical recycling reaching 25%.

[Issue 21] Supplementary information - Line 779: text to explain.

[Response] Accepted. We sincerely thank the reviewers for their insightful suggestion, which prompted us to add explanatory text to better emphasize to the reader the strategic focus of this scenario.

- **Supplemental Information, below Section 8.13 (Collect and dispose):** The program focuses on waste collection (Section 8.6) and disposal management (Sections 8.9-8.10) and assesses their overall contribution to synergistic abatement potential while prioritizing leakage control. Under baseline assumptions maintaining constant scenario weightings during individual variable modifications, key targets include: (1) mismanaged waste incidence reduced to 1%; (2) incineration achieving 50%; and (3) landfilling limited to 19%.

Itemized response to each review comment

“Strategies for synergistic reduction of plastic leakage and greenhouse gas emissions in China”

(NCOMMS-25-23856A)

On behalf of my co-authors, we thank you very much for allowing us to revise our manuscript. We would like to express our sincere appreciation for your careful reading and the reviewers’ insightful comments concerning our manuscript. All the review comments have been adequately addressed and clearly explained in the revised version of the manuscript. All the changes in the manuscript have been shown with yellow highlighting. Here is the itemized response to each comment with a specific statement, clearly addressing each issue and explaining what, why and how of the corresponding revision.

Reviewer # 1

[Issue 1] The manuscript presents valuable analytical findings; however, a significant weakness lies in the disconnection between these results and the policy implications section. The current implications are generic and not uniquely derived from the study's specific contributions. To strengthen the manuscript's impact, the authors must thoroughly reorganize this section to build a direct and actionable bridge from their results to concrete policy recommendations. Namely, authors should explicitly connect their results to actionable policy pathways. For instance, how can your current research findings be translated into practical applications to assist the government in managing plastics and carbon, as well as contribute to the development of an international plastics treaty? Does this study offer any enlightening insights for the management of international policies? How can these management policies be integrated with China's carbon neutrality goals to achieve synergistic emission reductions?

[Response] Accepted. We sincerely thank the reviewer for this insightful and constructive comment. We fully agree that the previous version of our Policy Implications section did not sufficiently reflect the unique contributions of our analysis and did not adequately connect our results to concrete

policy pathways. Following the reviewer's valuable suggestions, we have thoroughly reorganized the Discussion section to explicitly bridge our key findings with actionable and evidence-based policy insights at both national and international levels.

- **We have made the following major improvements:**

- 1. A clear linkage between analytical results and policy recommendations**

We directly connect each life-cycle hotspot identified in our analysis with its corresponding mitigation pathway, ensuring that all policy implications are now supported by quantitative evidence.

- 2. Mechanism-based explanation of why systemic interventions are required**

We added reasoning showing why single interventions are insufficient, based on the contrasting drivers of upstream GHG emissions and downstream leakage. This creates a clear causal bridge between evidence and policy design.

- 3. Polymer- and sector-specific mitigation strategies**

Building on our polymer-resolved analysis, we now highlight differentiated strategies. These are key intervention points uniquely revealed by our detailed material-flow model.

- 4. Integration with China's carbon-neutrality strategy and global treaty negotiations**

We explicitly discuss how our findings inform synergistic plastic-carbon governance in China and how our polymer-carbon coupled framework can support priority-setting, monitoring, and differentiated implementation under an international plastics agreement.

- 5. Clarification of the novelty and practical relevance of our framework**

We highlight that our polymer- and sector-resolved system evaluation provides the first detailed quantification of coupled emission-leakage patterns in China's plastics system, enabling policy insights that cannot be generated from aggregated assessments.

Collectively, these revisions create a strong and direct connection between our analytical findings and their policy implications. We believe that the revised Discussion now provides a much clearer, deeper, and action-oriented interpretation of our results, fully addressing the reviewer's concerns.

- **Lines 314-367:**

By integrating polymer-resolved material flow analysis with environmental leakage and GHG accounting, this study resolves the heterogeneous environmental profiles of plastics across life-cycle stages. Production, driven by fossil feedstocks and energy-intensive polymerization, dominates

plastics-related GHG emissions²⁴, whereas end-of-life mismanagement governs environmental leakage^{2,19,42}. Five polymers (PET, PP, LDPE, HDPE, and PVC) account for the majority of both impacts, indicating that environmental burdens are concentrated in specific material types and socioeconomic uses. These findings show that polymer properties and sectoral applications jointly determine environmental outcomes and underscore the need for differentiated mitigation strategies.

Scenario analysis indicates that single measures such as enhanced recycling or energy decarbonization are insufficient to generate substantial co-benefits. A fundamental mismatch exists between the drivers of upstream GHG emissions and the mechanisms of downstream plastic leakage. The system-change scenario integrates material reduction, design optimization, improved collection and recycling, and energy decarbonization, thereby reducing virgin demand at the source while lowering losses at end-of-life. By 2060, environmental leakage and net GHG emissions decline by roughly 80% and 63%, respectively. These results demonstrate that only coordinated, life-cycle-based governance can effectively deliver joint reductions in plastic pollution and climate impacts.

Differences across polymers and sectors further highlight the need for targeted mitigation. Plastic leakage is highly concentrated in a small set of critical material–sector combinations (Supplementary Tables 34–37). In aquatic systems, the dominant contributors are HDPE, LDPE, PP and textile PET; in terrestrial systems, leakage is primarily associated with packaging plastics (HDPE, LDPE, PP, PET), agricultural LDPE films and textile PET. These polymers exhibit both low circularity and high leakage risks, and their persistent dominance over time reveals structural weaknesses in product design, collection systems and end-of-life management. Targeted interventions for these high-risk polymers can generate disproportionate environmental benefits, align with carbon-neutrality and circular-economy objectives, and provide actionable pathways for differentiated commitments under an international plastics treaty.

China's in-use plastic carbon stock reached 1107.0 Mt CO₂-eq in 2021 and is projected to increase to 2485.0 Mt CO₂-eq by 2060 under the system-transition scenario (Supplementary Fig. 25). This growing stock represents an increasingly significant pool of embedded carbon whose long-term accumulation and eventual release depend on product lifetimes, material substitution decisions

and the efficiency of recycling loops. As these dynamics are not captured by production-based accounting alone, integrating plastics management into national energy-transition strategies, circular manufacturing systems and carbon-accounting frameworks is essential^{25,43–45}. Such integration would allow China to track embodied-carbon dynamics more accurately, avoid future stock-driven emissions and strengthen synergies between plastic governance and carbon-neutrality goals.

Overall, plastic mitigation requires differentiated and coordinated life-cycle strategies informed by polymer properties and sectoral applications. The polymer-resolved framework developed here provides the first quantitative evidence of the coupled emission–leakage characteristics of China’s plastic system, offering a scientific basis for synergistic management of plastic pollution and climate impacts. This framework can support global agreements in priority-setting, monitoring and differentiated implementation by identifying high-impact polymers, critical stages and optimal intervention pathways. Nationally, prioritizing high-contribution polymers, improving end-of-life collection and integrating circular material substitution, industrial decarbonization and carbon-stock accounting are expected to deliver the greatest combined benefits.

Manuscript References

2. Cottom, J. W., Cook, E. & Velis, C. A. A local-to-global emissions inventory of macroplastic pollution. *Nature* **633**, 101–108 (2024).
19. Lau, W. W. Y. *et al.* Evaluating scenarios toward zero plastic pollution. *Science* **369**, 1455–1461 (2020).
24. Stegmann, P., Daioglou, V., Londo, M., van Vuuren, D. P. & Junginger, M. Plastic futures and their CO₂ emissions. *Nature* **612**, 272–276 (2022).
25. Zheng, J. & Suh, S. Strategies to reduce the global carbon footprint of plastics. *Nat. Clim. Change* **9**, 374–378 (2019).
42. Pottinger, A. S. *et al.* Pathways to reduce global plastic waste mismanagement and greenhouse gas emissions by 2050. *Science* **386**, 1168–1173 (2024).
43. Bauer, F. *et al.* Plastics and climate change—Breaking carbon lock-ins through three mitigation pathways. *One Earth* **5**, 361–376 (2022).

44.Meys, R. *et al.* Achieving net-zero greenhouse gas emission plastics by a circular carbon economy. *Science* **374**, 71–76 (2021).

45.Suh, S. & Bardow, A. Plastics can be a carbon sink but only under stringent conditions. *Nature* **612**, 214–215 (2022).

Reviewer # 3

[Issue 1] Thank you for the careful review of the manuscript.

[Response] **Acknowledged.** We sincerely thank the reviewer for the careful and thoughtful evaluation of our manuscript. We appreciate the time and effort invested in reviewing our work.

General comments:

This study provides valuable insights into the dual challenges of environmental leakage and greenhouse gas (GHG) emissions faced by the plastics industry. By conducting a systematic analysis of the material metabolism, environmental leakage, and GHG emissions associated with 14 types of plastics in China from 1992 to 2021, and by modeling the synergistic emission reduction potentials and relative cost-effectiveness across 14 scenarios from 2021 to 2060, the research offers a comprehensive framework for understanding the complexities of plastic-related environmental impacts. The findings provide a critical reference for the development of targeted mitigation strategies tailored to specific plastic types and industrial sectors. However, several aspects of the manuscript could benefit from further refinement to enhance its clarity, depth, and broader impact.

Specific Comments:

1. The overall logic of the introduction is relatively clear, but the ending appears somewhat abrupt and could be improved. It is suggested to include a supplementary table comparing this study with previous research. A detailed comparison of methodologies, data sources, and conclusions could highlight the improvements in this study and describe how they were achieved.
2. The fonts used in the figures should be standardized to ensure consistency. For example, in Figures 2 and 4, the font style makes certain words hard to read due to letters being too close together.
3. In Figure 2a, should the final label on the x-axis be revised to "2021" for

consistency? Additionally, are the x-axis labels missing in subfigures d, e, and f?

4. What is the significance of the small section on the right side of Figure 3a? Including a corresponding annotation or note would enhance the clarity of the figure.
5. What is the calculation method and basis for the step from chemical feedstock to primary plastics? Please provide a more detailed explanation.
6. In the section on plastic product production and use, could a table be added to clarify the conversion from plastic resins to primary products? The table could also match these products to the corresponding production sectors, listing the proportion of different products flowing into each sector, along with the source of the conversion coefficients.
7. While the paper provides the composition ratios of various plastic resins in sectoral plastic products, it remains unclear how these data are used to calculate the volume of primary plastic products entering each sector.
8. Please review all formulas in the paper. Some formulas, such as Formula (6) in the main text and Formulas (10) and (11) in the appendix, contain variables or symbols not clearly linked to their explanations.
9. Several coefficients used in the calculations, along with their sources, are not clearly explained. It is recommended to include a supplementary table listing these coefficients—for example, the coefficients for plastic leakage to the environment, and emission factors for greenhouse gases during recycling, incineration, and landfilling.

10. In supplementary formula (20), how is $Oil_{k,t}$ calculated? Also, should the references to Table S11 and Table S12 in the explanation be corrected to Table S12 and Table S13? Please check for similar errors throughout the text.